# An improved Freezing Ice Nucleation Detection Analyzer (FINDA) for droplet immersion freezing measurement

Kaiqi Wang<sup>1,2,3</sup>, Kai Bi<sup>1,11,\*</sup>, Shuling Chen<sup>2,3,4</sup>, Markus Hartmann<sup>5</sup>, Zhijun Wu<sup>6</sup>, Jiyu Gao<sup>7</sup>, Xiaoyu Xu<sup>2,3,4</sup>, Yuhang Cheng<sup>2,3,4</sup>, Mengyu Huang<sup>1,11</sup>, Yunbo Chen<sup>1</sup>, Huiwen Xue<sup>6</sup>, Bingbing Wang<sup>7</sup>, Yaqiong Hu<sup>8</sup>, Xiongying Zhang<sup>9</sup>, Xincheng Ma<sup>1</sup>, Ruijie Li<sup>1</sup>, Ping Tian<sup>1</sup>, Ottmar Möhler<sup>8</sup>, Heike Wex<sup>5</sup>, Frank Stratmann<sup>5</sup>, Jie Chen<sup>10,\*</sup>, Xianda Gong<sup>2,3,\*</sup>

Correspondence to: Xianda Gong (gongxianda@westlake.edu.cn), Kai Bi (bikai\_picard@vip.sina.com), Jie Chen (jie.chen@env.ethz.ch)

Abstract. Heterogeneous ice nucleation initiated by atmospheric ice-nucleating particles (INPs) is a key microphysical process for cloud formation. Detecting the ice nucleation ability (INA) and concentration of INPs is essential for improving global climate models. Droplet freezing techniques (DFTs) are among the widely used tools for measuring the immersion freezing of INPs, which is a predominant ice nucleation process in mixed-phase clouds. To enhance the efficiency and accuracy of DFTs, we developed a Freezing Ice Nucleation Detection Analyzer at Westlake University (FINDA-WLU) with an improved hardware setup, user-friendly software, precise droplet freezing detection, and rigorous temperature calibrations. The temperature uncertainty of FINDA-WLU is about ±0.60 °C, considering both vertical heat transfer efficiency and horizontal temperature heterogeneity. The system is tested with Milli-Q ultrapure water and reference materials like Arizona Test Dust and Snomax®, and the results are consistent with previous studies. We also use the FINDA-WLU to measure INPs in precipitation samples collected in China. Overall, FINDA-WLU proved to be a reliable and precise method for measuring INA and INP concentrations.

<sup>&</sup>lt;sup>1</sup>Beijing Weather Modification Center, Beijing Meteorological Service, Beijing, China

<sup>&</sup>lt;sup>2</sup>Research Center for Industries of the Future, Westlake University, Hangzhou, 310030, China

<sup>&</sup>lt;sup>3</sup>Key Laboratory of Coastal Environment and Resources of Zhejiang Province, School of Engineering, Westlake University, Hangzhou, 310030, China

<sup>&</sup>lt;sup>4</sup>College of Environmental and Resource Sciences, Zhejiang University, Hangzhou, 310058, China

<sup>&</sup>lt;sup>5</sup>Leibniz Institute for Tropospheric Research, Leipzig, 04318, Germany

<sup>&</sup>lt;sup>6</sup>State Key Joint Laboratory of Environmental Simulation and Pollution Control, College of Environmental Sciences and Engineering, Peking University, Beijing, 100871, China

<sup>&</sup>lt;sup>7</sup>College of Ocean and Earth Sciences, State Key Laboratory of Marine Environmental Science, Xiamen University, Xiamen, 361102, China

<sup>&</sup>lt;sup>8</sup> Institute of Meteorology and Climate Research, Karlsruhe Institute of Technology, Karlsruhe, Germany

<sup>&</sup>lt;sup>9</sup>School of Earth Sciences, Yunnan University, Kunming, 650504, China

<sup>&</sup>lt;sup>10</sup>Institute for Atmospheric and Climate Science, ETH Zürich, Zurich, 8092, Switzerland

<sup>20 &</sup>lt;sup>11</sup>Field Experiment Base of Cloud and Precipitation Research in North China, China Meteorological Administration, Beijing, 101200, China

## 35 1 Introduction

50

Ice formation is one of the most important atmospheric processes to modulate cloud microphysics, thereby influencing precipitation and cloud radiative properties (Prenni et al., 2007; Bangert et al., 2012; Vergara-Temprado et al., 2018). Atmospheric ice formation can occur through homogeneous freezing of droplets or heterogeneous ice nucleation of liquid water or water vapor aided by aerosol particles, known as ice-nucleating particles (INPs). The homogeneous freezing of droplets, without the presence of INPs, usually occurs at temperatures (T) below -38 °C (Murray et al., 2012). In contrast, heterogeneous ice nucleation can occur at higher temperatures above -38 °C, because an INP provides a substrate which lowers the energy barrier required for ice embryo formation (Hoose and Möhler, 2012; Murray et al., 2012). Heterogeneous ice formation can occur through different modes, depending on environmental conditions and ice nucleation properties of INPs (Hoose and Möhler, 2012). Immersion freezing, where droplet freezing is initiated by an INP immersed within a droplet, is one of the most important pathways of heterogeneous ice nucleation in mixed-phase clouds (T > -38 °C) (Ansmann et al., 2008; Hiranuma et al., 2015; Westbrook and Illingworth, 2013). The concentration, physicochemical nature, and ice nucleation mechanisms of INPs are not fully understood, due to the stochastic nature of ice nucleation, low number concentration in the atmosphere, and the inherent complexity of INPs (Kanji et al., 2017).

Various measurement techniques have been developed and continuously upgraded over the past few decades to detect the immersion-freezing abilities of aerosol particles. These techniques include in-situ methods, including ice nucleation chambers and laminar flow reactors (Rogers, 1988; Rogers et al., 2001; Demott et al., 2015; Garimella et al., 2016; Lacher et al., 2017; Kanji et al., 2013; Möhler et al., 2021b), and offline droplet freezing techniques (DFTs) (Hill et al., 2014; Budke and Koop, 2015; Chen et al., 2018a; Miller et al., 2021; Chen et al., 2018b; Harrison et al., 2018; David et al., 2019; Steinke et al., 2020; Conen et al., 2015; Hill et al., 2016; Knackstedt et al., 2018; Gong et al., 2020; Mahant et al., 2023; Sze et al., 2023). Ice nucleation chambers and flow reactors are used to measure the ice formation potential of airborne particles under controlled T and relative humidity (RH) conditions. In chambers, aerosol particles are kept suspended, allowing aerosol particles to be activated into supercooled droplets and ice crystals and detected in situ. To measure the immersion freezing of droplets containing INPs, ice nucleation chambers are operated under mixed-phase cloud-relevant conditions, with T above – 38 °C and RH with respect to water at ~100%. The continuous flow diffusion chambers (CFDCs) (Demott et al., 2017; Lacher et al., 2017; Demott et al., 2018; Brunner and Kanji, 2021) and cloud expansion chambers (Möhler et al., 2021b; Möhler et al., 2021a) are two types of ice nucleation chambers operating on different working principles. However, ice nucleation chambers and reactors are typically expensive and have higher detection limits compared to DFTs. This enables them to measure often only at lower temperatures (T < -20 °C), particularly for typical atmospheric INP concentrations. Background noise caused by ice residues falling from chamber walls (e.g., CFDCs) or counting statistics of low ice crystal numbers make detecting INPs with low concentrations challenging (e.g., for both CFDCs and expansion chambers).

As an alternative, offline DFTs have been developed to measure the temperature-dependent freezing abilities of droplets containing aerosol particles. While different DFTs follow similar principles, the methods may differ for sample

collection, droplet preparation, and sample cooling (Hill et al., 2014; Budke and Koop, 2015; Chen et al., 2018a; Miller et al., 2021; Chen et al., 2018b; Harrison et al., 2018; David et al., 2019; Steinke et al., 2020). In DFTs, individual droplets with volumes ranging from microliter to picoliter are produced from aerosol water suspensions using pipettes or microfluidic chips liquid samples, and their freezing behavior is detected as the temperature is lowered. Common cooling methods include placing droplets within a cooling bath or onto a cold stage. Typically, the sampling time, droplet volume, and aerosol suspension concentration can be adjusted, which affects the particle number within each droplet and, thereby, its freezing ability. For example, particle numbers within a droplet can be enhanced by extending the aerosol sampling time, enlarging the droplet size, or reducing the dilution ratio of aerosol suspensions with water. In this way, this approach enables the quantification of low-concentration INP species in the atmosphere, which overcomes the high detection limitations of ice nucleation chambers. Due to these advantages, DFTs are widely used in current ice nucleation studies.

In this study, we present the newly developed Freezing Ice Nucleation Detector Array at Westlake University (FINDA-WLU), building on the original version of FINDA briefly introduced in Ren et al. (2024). The FINDA-WLU offers more precise temperature measurement and calibration, and frozen droplet detection (100% identification with an embedded automatic detection program). Details about the FINDA-WLU design, including the instrument setup and software control, are provided in Sect. 2. The temperature calibration and frozen droplet identification are explained in Sect. 3 and 4. The ice nucleation ability (INA) of Milli-Q ultrapure water, Arizona Test Dust (ATD), Snomax<sup>®</sup>, and precipitation samples are tested and compared with the results reported by previous studies using DFTs.

## 2 Instrument development

85

90

The FINDA-WLU instrument measures the INP number spectrum of liquid samples over a temperature range from 0.0 to about –30.0 °C with a changeable cooling rate of 0.1~1.0 °C. The measurement principle is based on the cold-stage freezing drop experiment, which ultimately yields INP number concentrations (Vali, 1971, 1994). Specifically, by monitoring the temperature-dependent frozen fraction of a large number of equally sized droplets and applying statistical analysis, the cumulative INP number spectrum of the original sample can be deduced. This method has been used in various atmospheric INP measurement instruments (Hill et al., 2014; Chen et al., 2018a).

## 2.1 Hardware setup

FINDA-WLU is based on the design of Hill et al. (2014), Schneider et al. (2021), and Ren et al. (2024), which was originally proposed by Vali (1971). As illustrated in Fig. 1a, FINDA-WLU comprises a custom-built aluminum block serving as a cold stage which is used to hold a 96-well Polymerase Chain Reaction (PCR) plate, a high-performance temperature-controlled refrigerated/heating circulator (HighTech FP50-HL, JULABO GmbH, Germany), four temperature sensors assembled to a data logger (NI 9217 and NI 9171, National Instruments Corporation, USA), a charge-coupled device (CCD) camera (GOX-5102-USB, JAI, Denmark), two LED lights (Luxpad23, Naguan, China), and a nitrogen gas purging system. All the

components are installed into a custom-built matte black metal cabinet mounted with two LED lights, which provides a stable environment light for image detection and reduces pollution from ambient air. A computer is connected to the CCD camera, temperature data logger, and the chiller's controller, running a customized LabVIEW program to operate FINDA-WLU and record the experiment data. A CCD camera (Fig. 1a) is used to detect the reflected LED light over the water droplets placed in a 96-well PCR plate during the experiment. This information can later be used to identify the freezing of droplets depending on experimental temperatures. The camera is fixed above the region of the PCR wells using an adjustable zoom lens (12-120 mm Focal Length, Qiyun Photoelectric Co., China).

An aluminum block is positioned within the coolant bath of the circulator and fitted with a silicone sealing strip to prevent contamination of the samples inside the aluminum block from direct contact with the coolant. Figures 1b and c show an image and a schematic of the aluminum block. An acrylic glass lid covers the aluminum block, providing insulation and preventing the air above the PCR plate from mixing with the surrounding air. A polytetrafluoroethylene (PTFE) component is sealed between the acrylic glass lid and the aluminum block, to decrease the heat transfer from the cold coolant to the acrylic glass lid to avoid fog and frost during the experiment (Fig. 1c). The aluminum block features a precisely Computer Numerical Control (CNC) machined cavity, designed to accommodate the 96-well PCR plate (0.2 ml PCR plate, Boibio, China) and four temperature sensors.

The four temperature sensors are platinum resistance thermometers Pt100 (5157701, YAGEO Nexensos, Germany), hereafter referred to as the 'Pt100 sensor' (Fig. 1b and c). These sensors are embedded and sealed within thermally conductive epoxy (Omegabond 200, Omega Engineering, Inc., USA) within tubes cut from a PCR plate, ensuring consistent heat transfer between the PCR plate and Pt100 sensors. The Pt100 sensors have an accuracy of ±0.15 °C at 0 °C (Class A, meets IEC 60751 standard). After calibration, the temperature of droplets was derived from the calibrated mean value of four sensors. Using the circulator, the maximum average cooling rate is –1.45 °C min<sup>-1</sup>, cooling the droplets from 0.0 °C to –35.0 °C. The instrument typically operates at a cooling rate of –1.0 °C min <sup>-1</sup> for experiments. In a standard experiment, the chiller is set to 0.0 °C and maintained for 10 minutes, then cooled at a rate of –1.0 °C min <sup>-1</sup> from 0.0 °C to –35.0 °C.

The purging system (as shown in Fig. 1a and b) blows clean and dry nitrogen at a flow rate of 6 L min<sup>-1</sup> over the surface of the PCR plate before the experiment to keep a clean atmosphere within the aluminum block. The clean and dry nitrogen is filtered by a 0.2 µm HEPA filter (Vent filters, HEPA-CAP, Whatman<sup>TM</sup>, United Kingdom) before being injected into the aluminum block. During the freezing experiment, the flow is halted to prevent heating of the air adjacent to the liquid sample in the PCR wells.

Figure 1: (a) The overview of the setup of FINDA-WLU. (b) The top view of the aluminum block with a PCR plate inside. (c) Schematic of the aluminum block from the front view.

#### 2.2 Software control

A customized National Instruments LabVIEW program was developed to control the experiment via a user interface panel shown in Fig. 2, including controlling the coolant bath circulator and monitoring the freezing status of droplets in the PCR plate with a CCD camera. The program consists of three primary interfaces: (1) sample information input (Fig. 2a), (2) droplet image acquisition and grayscale value extraction and analysis (Fig. 2b), and (3) temperature control and data logging (Fig. 2c). Before the cooling experiment starts, sample parameters that are needed to derive INP concentrations per e.g. volume of air, sample mass, etc. (see section 3) can be entered in the first interface (Fig. 2a). During the experiment, the CCD camera continuously monitors the unfrozen and frozen status of the liquid in the wells of the PCR plates. To enhance the freezing recognition accuracy, the zoom lens is adjusted to magnify the image so that the 96-well PCR plate occupies the entire display area. For accurate identification of each well, mask region parameters are manually defined in the software by the user (Fig. 2b); typically, circular masks with a diameter of 65 pixels arranged in an 8×12 array are employed. The software subsequently extracts and stores the grayscale values within these predefined mask regions. The program can analyze the temporal resolution of grayscale values and determine the frozen temperature of each droplet (details in Sect. 2.3). The frozen fraction and INP number concentration (for both water and air filter samples) are then calculated based on input sample information (calculation

methods in Sect. 4.5). The luminance data and four temperature readings are both recorded at a rate of 1 Hz. Simultaneously, temperature data from the PT100 sensors are recorded.

Figure 2: Screenshots of the software of FINDA-WLU. (a) Sample information. (b) The setting of grayscale recognition circles. The parameters are in the "Custom Setting" rectangle. The circles are shown in the right image. (c) The third page controls the cooling process and shows the real-time temperature of all Pt100 sensors. The image shown on the right side is the real-time CCD camera image. The numbers of "Realtime Image Pixel" are the grayscale values.

## 2.3 Droplet freezing detection

Immediate identification of the first ice formation in a droplet is crucial for quantifying the INA of immersion mode INPs. During the experiment, a freezing event is detected if the mean grayscale intensity of a well changes markedly (see Fig. 2b and c). The CCD camera acquires images at a frequency of 74 fps and provides a mean value of 1 Hz to the computer. This acquisition rate significantly exceeds previous studies, i.e., which captured 10 pictures per minute (Chen et al., 2018b). The increased image acquisition frequency improves the accuracy of the recognition of freezing temperature. The pixel intensities of each well area are averaged and recorded at a rate of 1 Hz. Figure 3 illustrates the grayscale values against the temperature of eight wells during an experiment. The grayscale value of a well stays constant until a sudden decrease is observed during a cooling experiment, indicating the onset of freezing. From 0.0 °C to -35.0 °C, the maximum decrease in grayscale value was used to identify the freezing event and the temperature at which it occurs.

Figure 3: The grayscale variation in an experiment. Black points are the recognized freezing points. The arrows refer to the grayscale image of this droplet at different phases.

#### 2.4 Temperature validation

The temperature of FINDA-WLU has been strictly calibrated with three steps: (1) calibrating the Pt100 sensors with a standard sensor, (2) calibrating an infrared camera with Pt100 sensors, and (3) calibrating single-well temperatures by using the calibrated infrared camera.

## 2.4.1 Temperature sensor (Pt100) calibration

The freezing temperature reported by the instrument is based on the mean value of four customized Pt100 sensors mounted near the corners of the PCR plate. A digital thermometer (Traceable<sup>®</sup>, Fisherbrand<sup>®</sup>, Thermo Fisher Scientific, USA; 0.001°C resolution) was calibrated in the temperature range relevant for the experiments by the National Institute of Metrology of China (the uncertainty is ±0.012 °C from 0.0 °C to -40.0 °C, Certificate No.: RGjc2023-04417) to serve as a reference. The calibrated digital thermometer is then used to calibrate the four Pt100 sensors. The calibrated digital thermometer and four Pt100 sensors are bound together during the calibration and immersed in the ethanol cooling bath at the same depth. The temperature within the bath is cooled down step-wise from 0.0 °C to -30.0 °C with an increment of -5.0 °C. To ensure uniform temperature distribution in the cooling bath, the chiller is held at each set temperature (0.0 °C, -5.0 °C, -10.0 °C, -15.0 °C, -20.0 °C, -25.0 °C, and -30.0 °C) for 20 minutes, and only the recorded data during the final 20 seconds are averaged and used for calibration. Figure 4 illustrates the temperature differences between the reference digital thermometer and four Pt100 sensors after calibration. The temperature differences are within ±0.02 °C. A linear regression was then applied to calibrate the temperatures of the four Pt100 sensors  $(T_1, T_2, T_3, \text{ and } T_4)$  using the calibrated digital thermometer's temperature as a reference. The results of the calibrated temperatures  $(T_{C1}, T_{C2}, T_{C3}, \text{ and } T_{C4})$  along with their coefficient of determination  $(r^2)$  are shown in Table 1. The standard deviation of this calibration is  $\pm 0.02$  °C (explained in Appendix B). The accuracy of Pt100 sensors is ±0.15 °C (mentioned in Sect. 2.1). The temperature uncertainty of Pt100 calibration is calculated as ±0.19 °C (explained in Appendix B). The mean of the four calibrated temperatures ( $T_{\text{C mean}}$ ) is calculated in Eq. (1) and used for further calibration.

$$T_{\text{mean}} = (T_{C1} + T_{C2} + T_{C3} + T_{C4})/4. \tag{1}$$

Table 1: The fitting to the data with the functional form  $T_{Ci} = a \times T_i + b$  (i = 1, 2, 3, 4)

| Sensor name       | a     | b     | $r^2$ |
|-------------------|-------|-------|-------|
| $T_{\rm C1}$      | 0.963 | 0.285 | 1.000 |
| $T_{\mathrm{C2}}$ | 0.964 | 0.297 | 1.000 |
| $T_{\mathrm{C3}}$ | 0.964 | 0.310 | 1.000 |
| <i>T</i> C4       | 0.963 | 0.234 | 1.000 |

Figure 4: The temperature differences between the four calibrated Pt100 sensors in FINDA-WLU and the reference digital thermometer after the calibration.

#### 2.4.2 Infrared camera calibration

The four calibrated Pt100 sensors are then used to calibrate the temperature measured by an infrared camera ( $T_{\rm Infrared}$ ), which is then used to measure the 96 wells' vertical and horizontal temperatures. The infrared camera has a resolution of 640×480 pixels and records the video thermography at 30 Hz. The infrared camera software extracts the temperature data, which is averaged to 1 Hz for analysis. The aluminum block is fixed and immersed in the ethanol of the refrigerated circulator, and a PCR plate is inserted into the aluminum block. The infrared camera is mounted above and focused on the center of an empty PCR well, and four Pt100 sensors are placed in the surrounding wells, as shown in Fig. 5a. The temperature within the bath was cooled down from 0.0 °C to -35.0 °C with an increment of -5.0 °C. Each temperature was held for 10 minutes, and the mean of  $T_{\rm Infrared}$  of the last 60 seconds was used for the calibration. As the accuracy of the infrared camera is not provided, we are using the variance during the last 60 seconds instead (details in Appendix B).

The mean temperature of four Pt100 sensors is considered to be equal to the temperature of the center well measured with the infrared camera and thus was used to calibrate the infrared camera. The functions of the calibrated infrared temperature  $(T_{C.Infrared})$  and  $r^2$  are:

$$T_{\text{C\_Infrared}} = 0.002 \times T_{\text{Infrared}}^2 + 0.873 \times T_{\text{Infrared}} + 5.059 \ (r^2 = 1.000).$$
 (2)

Figure 5b shows the infrared and the calibrated infrared temperature as red and black lines, respectively. The dashed gray line is the mean temperature of Pt100 sensors. The standard deviation of this quadratic regression calibration is  $\pm 0.09$  °C. The temperature uncertainty of infrared camera calibration is calculated as  $\pm 0.19$  °C (explained in Appendix B).

The calibration result is validated by observing the temperature of droplets in the PCR plate when they are freezing. When a water droplet freezes in the PCR well, the latent heat is released during the phase change and the droplet temperature increases to 0 °C (Harrison et al., 2018). We fill the PCR wells with 50 µL Milli-Q ultrapure water, and cool them down with

a cooling rate of 1.0 °C min<sup>-1</sup> from 0.0 °C to -35.0 °C. During the cooling,  $T_{\rm Infrared}$  of 20 randomly selected wells (white crosses in Fig. 5c.) is extracted with 1 Hz resolution. Figure 5d shows the temperature obtained by profiles of these 20 wells before ( $T_{\rm Infrared}$ ; blue lines) and after calibration ( $T_{\rm C\_Infrared}$ ; red lines). The mean temperature of the calibrated values at the ice-water equilibrium point during the experiment was close to 0.0 °C (-0.018±0.09°C), indicating that the infrared camera is well-calibrated.

Figure 5: (a) An image detected by the infrared camera during the calibration. The letters on the left and top show the labels of each well. (b) The temperature variation of the infrared camera before (red) and after (black) calibration during the freezing experiment.

The x-axis is the mean temperature of Pt100 sensors. The gray dashed line (mean temperature of Pt100 sensors) is a reference for the calibrated infrared temperature. (c) The infrared camera view for the validation experiment. The 20 selected PCR wells, where the temperature profile is extracted are marked with white crossings. (d) The temperature profiles of the 20 selected wells are measured by the infrared camera before (blue lines) and after (red lines) its calibration. The dashed gray line is a reference of 0.0 °C.

## 2.4.3 Vertical heat transfer

Simulation results indicate that the heat distribution throughout the aluminum block wells requires the precise placement of the temperature probe to ensure its readings accurately reflect the sample volume's temperature (Beall et al., 2017). We designed four freezing experiments to understand the heat transfer in the aluminum block, PCR plate, and the droplets in the FINDA-WLU. The four calibrated Pt100 sensors (see section 2.4.1) were placed in their four corner positions (the same as the regular freezing experiment) and the infrared camera was mounted above the aluminum block. These temperatures were measured in four different scenarios: (1) no PCR plate, T<sub>C\_Infrared</sub> measurement at the bottom of aluminum block wells; (2) with empty PCR plate, T<sub>C\_Infrared</sub> measurement at the bottom the PCR plate well; (3) with PCR plate filled with 50 μL dilliguity water per well, T<sub>C\_Infrared</sub> measurement at the liquid's surface in the well; (4) with PCR plate filled with 50 μL ethanol per well, T<sub>C\_Infrared</sub> measurement at the liquid's surface in the well. The temperature was decreased from 0.0 to –35.0 °C with a cooling rate of –1.0 °C min<sup>-1</sup> during these freezing experiments.

The differences between  $T_{\text{C,Infrared}}$  and the average temperature of four calibrated Pt100 sensors ( $T_{\text{C,mean}}$ ) as a function of chiller temperature is shown in Fig. 6. The similar temperature variation between the bottom of the aluminum block (red line) and the bottom of the empty PCR plate (cyan line) indicates a good heat transfer from the aluminum block to the PCR plate. A sudden increase in the surface temperature of water droplets (yellow line) indicates the release of latent heat upon freezing. After freezing, the temperature profile aligns closely with that observed at the bottom of the empty PCR plate, suggesting that heat conduction is efficient among the aluminum block, PCR plate, and water droplet at a cooling rate of  $-1.0~^{\circ}$ C min<sup>-1</sup>. Importantly, the temperature between the surface of the water droplet and the bottom of the empty PCR plate shows a large difference before the water droplet freezing, indicating that the heat transfer is delayed from the bottom of the PCR plate to the surface of the water droplet. Therefore, the infrared camera calibration should be performed in the empty PCR plate, as we have done in Sect. 2.4.2. A temperature discrepancy between water and anhydrous ethanol was also observed, which is likely attributable to differences in their heat capacities.

An additional experiment was performed to evaluate the temperature difference between the Pt100 sensors and the bottom of the aluminum block wells with a much lower cooling rate. We use a -5.0 °C temperature step from 0.0 °C to - 35.0 °C, with each temperature maintained for 30 minutes to allow for sufficient heat transfer. The different temperature variation between this experiment (black squares in Fig. 6) and the empty PCR plate indicates that the cooling rate affects the heat transfer, as well as the temperature calibration.

Figure 6: Temperature differences between the infrared camera and the mean temperature of the four calibrated Pt100 sensors as a function of the chiller temperature.

#### 2.4.4 Horizontal temperature calibration

The temperature bias across 96-well PCR plates has been discussed for aluminum block-based instruments with simulations (Beall et al., 2017), calibration substance freezing experiments (Kunert et al., 2018), and by comparison of temperature differences between corner and center wells (David et al., 2019). The horizontal temperature distribution at the bottom of an empty 96-well PCR plate is measured by the calibrated infrared camera ( $T_{C\_Infrared}$ ), together with four Pt100 sensors placed in the corners. The infrared camera recorded the temperature at 30 Hz and averaged the data to 1 Hz to fit the Pt100 data frequency. The freezing experiment was conducted from 0.0 °C to -35.0 °C at the rate of -1.0 °C min<sup>-1</sup>. Figure 7 shows the results of the temperature of each empty PCR well at 0.0 °C, -10.0 °C, -20.0 °C, and -30.0 °C (refer to  $T_{C\_mean}$ ). A clear heterogeneous temperature distribution existed. The PCR wells near the side boundary are warmer than those near the center, which was also observed in a previous study (David et al., 2019).

To address the horizontal temperature heterogeneity of the PCR plate, an individual-well calibration approach was conducted. The mean temperature of the four calibrated Pt100 sensors ( $T_{C\_mean}$ ) was used to calibrate the temperature of each PCR well:

$$T_{\text{C\_Infrared}_i} = a_i \times T_{\text{C\_mean}} + b_i \quad (i = 1, 2, 3, ..., 96),$$
 (3)

where  $T_{\text{C_Infrared_i}}$  is the calibrated temperature of the  $i^{\text{th}}$  well, and  $a_i$  and  $b_i$  are the slope and intercept of the regression, respectively. For each PCR well, a standard deviation of Eq. (3) is calculated, and two times the largest standard deviation ( $\pm 0.22$  °C) is treated as the uncertainty for this step. Overall, the temperature uncertainty of FINDA-WLU is  $\pm 0.60$  °C (explained in Appendix B).

Figure 7: The infrared temperature of each empty PCR well at (a) 0.0 °C, (b) -10.0 °C, (c) -20.0 °C, and (d) -30.0 °C.

#### 3 INP concentration calculation

The frozen fraction (FF) of droplets can be calculated based on Eq. 4, which assumes that droplet freezing is only temperaturedependent:

$$FF(T) = \frac{N(T)}{N_0},\tag{4}$$

where N(T) denotes the number of droplets that have frozen at temperature T, and  $N_0$  represents the total number of droplets.

Following Vali (1971), the cumulative number concentration of INPs ( $C_{INP}(T)$ ) per droplet, can be obtained based on Eq. (5):

$$C_{\rm INP}(T) = \frac{-\ln(1 - FF(T))}{V_{\rm droplet}},\tag{5}$$

where  $V_{\text{droplet}}$  represents the volume of the droplet, which is 50  $\mu$ L in our measurements.

For suspensions for which a certain mass of ice active material was added to a certain volume of water, the mass concentration is  $C_{\rm m}$ . With that,  $C_{\rm INP}(T)$  can be further normalized to estimate the INP numbers per unit mass of ice active material based on Eq. (6):

$$n_{\rm m} = \frac{C_{\rm INP}(T)}{C_{\rm m}}. (6)$$

 $C_{INP}(T)$  is calculated from statistical analysis; therefore, it is necessary to assess the reliability of the results. According to the binomial distribution method proposed by Agresti and Coull (1998), the 95% confidence interval of the FF at temperature T,  $CI_{95\%}(T)$ , is calculated as:

$$CI_{95\%}(T) = \frac{\frac{N(T)}{N_0} + \frac{1.96^2}{2N_0} \pm 1.96\sqrt{\frac{\left[\frac{N(T)}{N_0}\left(1 - \frac{N(T)}{N_0}\right) + \frac{1.96^2}{4N_0}\right]}{N_0}}}{1 + \frac{1.96^2}{N_0}}.$$
 (7)

## 285 4 Freezing experiments

The performance of FINDA-WLU was validated by testing the freezing behavior of Milli-Q ultrapure water, ATD, Snomax<sup>®</sup>, and precipitation samples and comparing them with previous studies.

#### 4.1 Sample preparation

Ultrapure water (resistivity > 18 M $\Omega$  cm and TOC < 10 ppb) (Ohmi, 2017) produced by the Milli-Q $^{\text{@}}$  IQ 7000 system was used in our experiments. Suspensions of Arizona Test Dust (ATD, Powder Technology, Inc.; ISO 12103-1, A4 Coarse) and Snomax $^{\text{@}}$  were prepared by dispersing the respective solid particles in ultrapure water. Both materials are widely used to evaluate the performance of DFT. The ATD suspension with a mass concentration of  $1.04\times10^{-1}$  g L $^{-1}$  was prepared by mixing 3.11 mg ATD with 30 mL ultrapure water. The ATD solid sample was weighted using a high-precision balance (Cubis MSA6.6S-0CE-DF, Sartorius, Germany), and the ultrapure water was then added into a 50 mL conical tube (Eppendorf, Germany). The suspension was then shaken on a shaking platform (NMSG-12, NuoMi $^{\text{@}}$ , China) for 20 minutes at its maximum rate to ensure homogeneity. The suspension was diluted to different concentrations to investigate the INA of ATD over a broader and lower temperature range (David et al., 2019). Suspensions with other concentrations, including  $1.04\times10^{-4}$ ,  $1.04\times10^{-3}$ , and  $1.04\times10^{-2}$  g L $^{-1}$ , were prepared by diluting the original  $1.04\times10^{-1}$  g L $^{-1}$  ATD suspension with ultrapure water. Similarly, a Snomax $^{\text{@}}$  suspension with a concentration of  $1.00\times10^{-1}$  g L $^{-1}$  was initially prepared by mixing 3.01 mg Snomax $^{\text{@}}$  with 30 mL ultrapure water. Since Snomax $^{\text{@}}$  shows INA at temperatures above -3.0 °C (Wex et al., 2015; Wieber et al., 2024), it must be diluted to a much lower concentration than ATD to observe its INA at low temperatures. The dilutions ranged from 10 to  $10^{\text{@}}$  fold, yielding concentrations of  $1.00\times10^{-8}$ ,  $10^{-7}$ ,  $10^{-6}$ ,  $10^{-5}$ ,  $10^{-4}$ ,  $10^{-3}$ , and  $10^{-2}$  g L $^{-1}$ .

In a cooling experiment, the suspension of ATD or Snomax<sup>®</sup> with one concentration was pipetted into a PCR plate using an electric pipette (Eppendorf Xplorer<sup>®</sup> plus, Variable 50-1000 $\mu$ L) to form 96 droplets, each with a volume of 50  $\mu$ L. Note that the suspension was re-shaken after pipetting every 32-drops to reduce the sedimentation of solid particles and reduce the difference in concentration of droplets.

Precipitation samples were collected in situ, with details on location, time duration, sample methods, and other relevant information provided in Tab. A1. All samples were stored in a freezer at -20 °C before analysis. The dilution was

prepared by first diluting 1 mL of the original sample with 14 mL of ultrapure water, creating a 15-fold dilution. It was then further diluted by mixing 1 mL with 14 mL of ultrapure water, producing a 225-fold dilution. The 3375-fold dilution was made similarly by mixing 1 mL of 225-fold dilution with 14 mL ultrapure water. Each dilution was mixed thoroughly before the next diluting operation.

#### 4.2 Freezing of droplets from Milli-Q water

The freezing behavior of ultrapure water used to prepare sample suspensions was first measured to quantify background INPs present in the ultrapure water. The measurement also serves as a quality control for the immersion freezing of sample droplets. Milli-Q water is tested and the FF as a function of temperature is shown in Fig. 8. The median freezing temperature ( $T_{50}$ ) of droplets, where 50% of droplets are frozen (FF=50%), is  $-26.5\pm0.04$  °C. The temperatures obtained by repeating experiments are similar, demonstrating the stable quality of the ultrapure water and the reproducibility of the freezing measurements.

The *FF* of Milli-Q water droplets using DFTs with different volumes, including Freezing Ice Nuclei Counter (FINC) (Miller et al., 2021), microtiter plate-based ice nucleation detection results in gallium (Micro-PINGUIN) (Wieber et al., 2024), Droplet Ice Nuclei Counter Zurich (DRINCZ) (David et al., 2019), InfraRed-Nucleation by Immersed Particles Instrument (IR-NIPI) (Harrison et al., 2018), and Ice Nucleation Droplet Array (INDA) (Chen et al., 2018b), are shown in Fig. 8 for comparison. In general, FINDA-WLU (*T*<sub>50</sub> = -26.5 ± 0.04°C) shows a considerably lower *T*<sub>50</sub> compared to those measured by INDA (*T*<sub>50</sub> = -25.5 °C), FINC (*T*<sub>50</sub> = -25.4 °C), DRINCZ (*T*<sub>50</sub> = -22.2 °C), IR-NIPI (*T*<sub>50</sub> = -21.0 °C), and Micro-PINGUIN (30 μL) are smaller than in FINDA-WLU, while the droplet size used in other DFTs, such as DRINCZ, is equal to the one in FINDA-WLU. The smaller droplets typically tend to freeze at lower temperatures than larger droplets due to the lower likelihood of forming an ice embryo in smaller droplet volumes. Therefore, the even lower *T*<sub>50</sub> of larger droplets obtained in our measurements compared to other studies, indicating a comparably low water background and less contamination introduced during our measurement.

Figure 8: Frozen fraction of Milli-Q water. The results of FINDA-WLU are shown as triangles, dots, and stars. The shaded area indicates the measurement uncertainties. Results for other droplet freezing techniques (micro-PINGUIN and DRINCZ) are shown as purple and green lines, respectively.

## 4.3 Ice nucleation of Arizona Test Dust

ATD has been widely used as a mineral dust proxy by ice nucleation communities (Bundke et al., 2008; Welti et al., 2009; Perkins et al., 2020; Mahant et al., 2023). The *FF* of ATD-containing droplets is shown in Fig. A1. The INP number per unit mass ( $n_{\rm m}$ ) of ATD is derived based on Eq. (3) and the results are shown in Fig. 9. The uncertainties associated with  $n_{\rm m}$  are indicated by the shaded area, which is derived according to the method suggested by Agresti and Coull (1998).  $n_{\rm m}$  of ATD increases with decreasing temperatures, as expected, with more INPs becoming ice-active at lower temperatures.  $n_{\rm m}$  obtained from droplets featuring different ATD concentrations (1.04×10<sup>-4</sup> to 1.04×10<sup>-1</sup> g L<sup>-1</sup>) align well with each other in the overlapping temperature ranges. This indicates that the  $n_{\rm m}$  of ATD scales with particle mass, which is also observed by many other studies using ATD (Murray et al., 2011; Niedermeier et al., 2011; Boose et al., 2016).

The  $n_{\rm m}$  of ATD measured by the Colorado State University ice spectrometer (CSU-IS) (Perkins et al., 2020) and a cold-stage developed by Mahant et al. (2023) is also shown in Fig. 9 for comparison. CSU-IS detected the freezing of 32 droplets with a volume of 50  $\mu$ L at a cooling rate of 0.3 °C min<sup>-1</sup>. The cold stage measured the freezing of 100 droplets with a volume of 1  $\mu$ L at a cooling rate of 4.0 °C min<sup>-1</sup> (Mahant et al., 2023). In general, the  $n_{\rm m}$  values of ATD measured in this study agree well with those reported by CSU-IS (Perkins et al., 2020); the difference is within one order of magnitude. At a given

temperature, our  $n_{\rm m}$  results are 2 to 13 times higher than those reported by Mahant et al. (2023). This difference could be attributed to the faster cooling rate employed by Mahant et al. (2023), which brings out the slight time dependence of the ice nucleation of ATD (Wright et al., 2013; Jakobsson et al., 2022).

Figure 9: The  $n_{\rm m}$  of ATD depending on temperature. The data obtained from our study are shown by colourful lines, and the corresponding shaded areas indicate the uncertainties of  $n_{\rm m}$ , derived by Agresti and Coull (1998). The results reported by CSU-IS measurements and by Mahant et al. (2023) are shown as gray strips and purple dots, respectively.

## 4.4 Ice nucleation of Snomax®

Snomax® is suggested as a reference ice-active material for immersion freezing experiments (Wex et al., 2015). Snomax® is the fragment of Pseudomonas syringae bacteria, which is widely used in generating artificial snow (Hiranuma et al., 2015; Whale et al., 2015). FINDA-WLU measured the temperature-dependent  $n_{\rm m}$  of Snomax®. The results in this study, together with those reported by previous studies (Budke and Koop, 2015; Wex et al., 2015; Tobo, 2016; Kunert et al., 2018; Steinke et al., 2020; Wieber et al., 2024) are shown in Fig. 10. Similar to ATD measurements,  $n_{\rm m}$  obtained from Snomax® droplets with different mass concentrations ( $1.04 \times 10^{-9}$  to  $1.04 \times 10^{-1}$  g L<sup>-1</sup>) align well with each other in the overlapping temperature range, indicating that INP numbers provided by Snomax® scale with its mass. In general,  $n_{\rm m}$  of Snomax® increases with decreasing temperature and reaches its maximum at -9.0 °C. A plateau appears in the  $n_{\rm m}$  spectra below -9.0 °C, indicating that a stronger supercooling will not further promote INP numbers per mass of Snomax below this temperature. This plateau is also observed by ice nucleation studies of Snomax® using other DFTs (Budke and Koop, 2015; Wex et al., 2015; Tobo, 2016; Kunert et al.,

2018; Steinke et al., 2020; Wieber et al., 2024). At temperatures below  $-7.0\,^{\circ}$ C,  $n_{\rm m}$  of Snomax® measured by FINDA-WLU is comparable to those reported by other DFTs; the difference is within one order of magnitude. However, at temperatures warmer than  $-7.0\,^{\circ}$ C,  $n_{\rm m}$  measured by FINDA-WLU are about two orders of magnitude lower than those detected using the Twin-plate Ice Nucleation Assay (TINA) (Kunert et al., 2018) and the Bielefeld Ice Nucleation Array (BINARY) (Budke and Koop, 2015). The steep increase of  $n_{\rm m}$  detected by TINA and BINARY at  $-4.0\,^{\circ}$ C to  $-7.0\,^{\circ}$ C indicates there is another type of INP in the respective Snomax® that can nucleate ice in this temperature range, which, however, was not detected in our study. Our results are consistent with those detected by the Ice Nucleation Spectrometer of the Karlsruhe Institute of Technology (INSEKT) (Steinke et al., 2020) and micro-PINGUIN, as they also did not observe such an INP type above  $-7.0\,^{\circ}$ C. The variation of Snomax®  $n_{\rm m}$  reported by different studies at temperatures warmer than  $-7.0\,^{\circ}$ C can be attributed to several reasons, as suggested by previous studies (Wieber et al., 2024; Tarn et al., 2018; Polen et al., 2016). For example, the ice nucleation abilities of Snomax® can be influenced by its storage time in the fridge and by aqueous aging when it is suspended in water (Wieber et al., 2024; Polen et al., 2016). Caution should be given when using Snomax® as a reference material to test and compare the performance of different instruments, as its INA is unstable, particularly at warm temperatures (>  $-7.0\,^{\circ}$ C).

Figure 10:  $n_{\rm m}$  of Snomax<sup>®</sup>. The colourful lines with shaded areas are  $n_{\rm m}$  and uncertainties are obtained by our measurements. The  $n_{\rm m}$  data reported by other studies (Wex et al., 2015) using different DFTs are shown for comparison.

## 4.5 INP numbers in precipitation samples

In addition to reference samples, we also tested atmospherically relevant samples to validate the FINDA-WLU performance. Four precipitation samples, including cloud water, snow water, hail, and typhoon water, are measured. The detailed sampling information is summarized in Tab. A1. Figure 11 shows  $C_{INP}(T)$  of the original suspensions of precipitations and three diluted suspensions (15-times, 225-times, 3375-times). As expected,  $C_{INP}(T)$  values of different dilution suspensions align well with the original suspensions in the overlapping temperature range. The  $C_{INP}(T)$  values of precipitation samples in this study are compared to those reported by Petters and Wright (2015), who summarized INP measurements from precipitation samples collected mostly over Europe and North America. The  $C_{INP}(T)$  values measured for our precipitation samples in general fall within the range reported by Petters and Wright (2015), except that the cloud water sample shows extremely high concentrations at -15.0 °C.

 $C_{INP}(T)$  of different samples shows a large variability. For example,  $C_{INP}(T)$  varies about three orders of magnitude, ranging from  $10^4$  L<sup>-1</sup> to  $10^7$  L<sup>-1</sup> at -14.0 °C among different samples. The cloud water shows the highest INP concentration, while the typhoon water has the lowest INP concentration. The large variation of INP concentration among different precipitation samples indicates that aerosol sources might be different or the aerosols underwent different aging and cloud processes. The high INP concentrations in cloud water, snow, and hail at warm temperatures (>-18.0 °C) suggest that biological aerosols might make a great contribution to INPs. Further chemical analysis and heating treatments of samples will help in the future to confirm the nature and sources of INPs.

Figure 11: The INP number concentration ( $C_{INP}(T)$ ) of four precipitations sample in different colourful lines. The gray dashed area indicates the INP results of precipitation samples collected globally, as reported by Petters and Wright (2015). The numbers in the brackets refer to the dilution factor of the samples. "1" means the original liquid of the samples. "15", "225", and "3375" means the 15-time, 225-time, and 3375-time fold dilutions.

## **5 Summary**

In this study, we present an improved Freezing Ice Nucleation Detection Analyzer developed at Westlake University (FINDAWLU) for immersion mode INP measurements. We designed a robust hardware setup and user-friendly software working together seamlessly, ensuring that the instrument operates smoothly and efficiently. We take into account the vertical heat transfer from the cooling bath to the aluminum block and the PCR plate, as well as the temperature heterogeneity across the PCR wells, for temperature calibration. The temperature calibration includes three steps: (1) calibrating the Pt100 sensors with a standard sensor, (2) calibrating an infrared camera with Pt100 sensors, and (3) calibrating single-well temperatures using an infrared camera. For the freezing experiment at the temperature range from 0.0 to -30.0 °C with a cooling rate of -1.0 °C min<sup>-1</sup>, the temperature uncertainty is  $\pm 0.60 \text{ °C}$ .

INP concentrations of ultrapure water detected by FINDA-WLU are lower than reported in one previous study and consistent with another. Two ice active substances often used in laboratory studies, ATD and Snomax<sup>®</sup>, are also tested and showed similar results to previous studies. Both tests prove the good performance of FINDA-WLU. The precipitation samples

collected in China show a large variation of INP concentrations, which are, however, within the range reported by previous studies. In conclusion, the FINDA-WLU has a very thorough temperature calibration among INP offline immersion freezing instruments. It has an enhanced temperature accuracy, and it also has an increased measurement efficiency with its user-friendly software.

Appendix A: Frozen fraction (FF) results and information about different samples

Figure A1: The FF of ATD and Milli-Q water. The dashed areas are the uncertainties of FF.

Figure A2: The FF of Snomax® and Milli-Q water. The dashed areas are the uncertainties of FF.

Figure A3: The FF of the Milli-Q water and precipitation. The dashed areas are the uncertainties of FF. The numbers in the brackets refer to the dilution factor of the samples. "1" means the original sample. "15", "225", and "3375" means the 15-time, 225-time, and 3375-time fold dilutions.

Table A1: Information on four precipitation samples, including cloud water, snow, hail, and typhoon water. All samples preserved at -20 °C in a fridge.

| Precipitation type | Location (latitude, longitude, sea level height) | Local time (UTC+8)                        | Information                                                                                                                                                |
|--------------------|--------------------------------------------------|-------------------------------------------|------------------------------------------------------------------------------------------------------------------------------------------------------------|
| Cloud water        | 35.66° N, 106.20° E, 2642 m                      | 2023-08-06, 21:00 to<br>2023-08-07, 01:00 | A portable passive cloud water collector on the Liupan mountain collected the cloud water for 4 hours. The total volume of collected cloud water is 45 mL. |
| Snow               | 40.52° N, 115.78° E, 1314 m                      | 2023-12-14, 19:00 to 20:00                | A PTFE collecting box was placed on the Haituo Mountain for one hour.                                                                                      |
| Hail               | 39.94° N, 116.29° E, 56 m                        | 2021-06-30, 19:00                         | A hail sampler was put on the ground in Beijing during a hail event. The hail was then separated from the rain.                                            |

**Typhoon** 39.94° N, 116.29° E, 56 m 2023-07-30, 15:00 water

A PTFE collecting box was placed on the ground in Beijing during a typhoon event. The total volume of collected typhoon water is 50 mL.

## Appendix B: Calculation of uncertainty

## The uncertainties of simple linear and quadratic regressions

For a pair of data lists x and y, x are used to calibrate y by linear regression. a is the slope of the calibration function, and b is the intercept. For each known  $x_i$ , the calibrated value  $\hat{y}_i$  is:

$$\hat{y}_i = a \cdot x_i + b \tag{B1}$$

In the quadratic regression, for each known  $x_i$ , the calibrated value  $\hat{y}_i$  is:

$$\hat{y}_i = a \cdot x_i^2 + b \cdot x_i + c \tag{B2}$$

where a and b are the coefficients, and c is the constant.

According to Sharaf et al. (1986), Miller (1991), and Hibbert (2006), the standard deviation of the regression  $sd_r$  (a.k.a. the uncertainty of the calibration) is calculated as:

$$sd_{\rm r} = \sqrt{\frac{\sum_{i=1}^{n} (y_i - \hat{y}_i)^2}{n-k}},$$
 (B3)

where n is the total number of the x or y, i represents the sequence number of x or y, and k is the number of coefficients in the regression model.

## The uncertainty of FINDA-WLU

We have three calibration steps for FINDA-WLU. According to Eq. B3, the standard deviations of four Pt100 sensors are  $sd_{p1}$ ,  $sd_{p2}$ ,  $sd_{p3}$ , and  $sd_{p4}$ , respectively (Sect. 3.1). The accuracy of the Pt100 sensor is  $\pm 0.15$  °C (written as  $e_1$ ). According to Ku (1966) and Neuhauser and Roper (2004), the uncertainty of the Pt100 sensor (the first step) can be written as

$$u_{\text{step1}} = 2 \times \sqrt{sd_{\text{p1}}^2 + sd_{\text{p2}}^2 + sd_{\text{p3}}^2 + sd_{\text{p4}}^2} + e_1,$$
 (B4)

where the  $e_1$  is the accuracy of the Pt100 sensors. Here we consider the two standard deviations, which cover >95% of data points (assuming normal distribution).

In the second step, we first measured the accuracy of an infrared camera. The bath temperature was cooled down from 0.0 to -35.0 °C, with an increment of -5.0 °C. Each temperature was held for 10 minutes, and the last 60 seconds were used for the accuracy calculation. There are 8 (k = 8) groups of temperature, and 60 (n = 60) data in each group. Therefore, the accuracy ( $e_2$ ) of the infrared camera is calculated as:

$$e_2 = \sqrt{\frac{\sum_{i=1}^k \sum_{j=1}^n (x_{ij} - \bar{x}_i)^2}{k(n-1)}}.$$
 (B5)

The standard deviation of the quadratic regression Eq. (2) is  $sd_{IR}$ . The uncertainty of the infrared camera calibration  $u_{\text{step2}}$  is:

$$u_{\text{step2}} = 2 \times sd_{\text{IR}} + e_2. \tag{B5}$$

In the third step, we considered the uncertainties of the horizontal temperature calibration. Each well has a standard deviation  $(sd_{s,i})$  and the maximum standard deviation value of the single-well is selected, as:

$$sd_s = maximum(sd_{s,i}) \quad (i = 1, 2, ..., 96).$$
 (B6)

Therefore, when considering the two standard deviations, the uncertainty of the third calibration is:

$$u_{\text{step3}} = 2 \times sd_{\text{s}}.\tag{B7}$$

The total uncertainty of FINDA-WLU is calculated as:

$$u = u_{\text{step1}} + u_{\text{step2}} + u_{\text{step3}}.$$
 (B8)

The uncertainties are summarized in Tab. B1.

Table B1: The calculation of temperature uncertainties

| Calibration          | Accuracy                        | Standard deviation                                                      | Uncertainties               |
|----------------------|---------------------------------|-------------------------------------------------------------------------|-----------------------------|
| Step1: Pt100 sensors | <i>e</i> <sub>1</sub> =±0.15 °C | $sd_{p} = \sqrt{sd_{p1}^{2} + sd_{p2}^{2} + sd_{p3}^{2} + sd_{p4}^{2}}$ | $2 \times sd_{\rm p} + e_2$ |
| $(u_{step1})$        |                                 |                                                                         | =±0.19 °C                   |

|                        |                 | =±0.02 °C                |                                                          |
|------------------------|-----------------|--------------------------|----------------------------------------------------------|
| Step2: Infrared camera | $e_2$ =±0.01 °C | $sd_{\rm IR}$ = ±0.09 °C | $2 \times sd_{1R} + e_2$                                 |
| $(u_{step2})$          |                 |                          | = ±0.19 °C                                               |
| Step3: Sigle well      | NA              | $sd_s$ = ±0.11 °C        | $2 \times sd_s$                                          |
| $(u_{step3})$          |                 |                          | = ±0.22 °C                                               |
| Total                  |                 |                          | $u_{\text{step1}} + u_{\text{step1}} + u_{\text{step1}}$ |
|                        |                 |                          | =±0.60 °C                                                |

Data availability. All data are available upon request from the corresponding authors.

Author contributions. X. G., K. B., and J. C. conceived and designed the study, with input from all other authors.

Competing interests. The authors declare that they have no conflict of interest.

Acknowledgements. K.B. and Y.C. are supported by the Beijing National Science Foundation (8244065). K. B., P. T., and X. G. are supported by the National Natural Science Foundation of China (42275087, 42105091, 41930968, 22576164, 42505070). X. G., X. X., Y. C., and S. C. are supported by the Opening Project of Shanghai Key Laboratory of Atmospheric Particle Pollution and Prevention (LAP3) (No. FDLAP24018), the Research Center for Industries of the Future (RCIF) at Westlake University (No. 103110746022301, 210000006022303), and Westlake Education Foundation. J. C. acknowledges support from the Swiss National Science Foundation (SNSF) postdoctoral fellowship (No. TMPFP2\_209830). M.H. and R.L. are supported by the Beijing Municipal Science and Technology Commission (Z251100004525005) and the Innovation Project of the China Meteorological Administration (CXFZ2025J151). The financial support by the China Scholarship Council (CSC) for K. B. and Y. H. is also gratefully acknowledged. J. G. and B. W. are supported by the National Natural Science Foundation of China (42375069). We thank Zhiliang Shu and Lei Tian from Liupan Mountains Atmospheric Science Field Observation and Research Station in Ningxia Hui Autonomous Region for their help in collecting cloud water samples.

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
