# Peer review of "An improved Freezing Ice Nucleation Detection Analyzer (FINDA) for droplet immersion freezing measurement"

_EGUsphere, 2025_

## Author Comment (AC1)

**Reviewer #1**

This manuscript presents an advancement in droplet freezing techniques (DFTs) for measuring ice-nucleating particles (INPs) via immersion freezing. The development of FINDA-WLU addresses uncertainties in temperature control, detection accuracy, and operational efficiency. While the study is methodologically sound and provides validation data, several aspects require clarification to establish the novelty and reliability of the instrument. Below are detailed comments and suggestions for improvement.

We would like to thank the reviewers for their thoughtful comments that helped improve our manuscript. We revised the manuscript accordingly and think it has strengthened as a result. Please find our point-by-point response in blue text. Additions to the text are shown in *italics with an underline*. All line numbers refer to the new version of the manuscript. A tracked changes version is also included.

The authors claim improvements in hardware, software, and temperature calibration, but the specific innovations need explicit articulation. Compared to prior FINDA designs, FINDA-WLU achieves ±0.60°C uncertainty. However, the manuscript should clarify how the heat transfer efficiency (vertical) and temperature homogeneity (horizontal) were optimized.

Thank you for your suggestions. Compared to the original version of FINDA (Ren et al., 2024), the FINDA-WLU has undergone structural optimization, ensuring a more secure fixation of components such as the CCD and LED lights. Most importantly, the design of the core element, the aluminum block cold stage, has been re-engineered to enhance both performance and stability. In the previous version, four Pt100 sensors were attached to the inner bottom of the four corner wells of the 96-well plate, and these wells were then fixed to the cold stage. In FINDA-WLU, the wells designed to accommodate the four Pt100 sensors have been repositioned to the outer edge of the 96-well plate, eliminating the need to cut the four corners of the PCR plate before each experiment.

Regarding the software, the original version of FINDA used three separate software packages to control the chiller, read the Pt100 data, and acquire the CCD image. In contrast, FINDA-WLU combines all these functions into a single software package. The updated software in FINDA-WLU also includes automated grayscale analysis and calculation of INP number concentrations. We added the following text in Lines 140 to 144:

*"The program can analyze the temporal resolution of grayscale values and determine the frozen temperature of each droplet (details in Sect. 2.3). The frozen fraction and INP number concentration (for both water and air filter samples) are then calculated based on input sample information (calculation methods in Sect. 4.5)."*

Regarding temperature homogeneity, FINDA-WLU includes single-well temperature calibration, as explained in Sect. 2.4.4, Horizontal Temperature Calibration.

Since the original version of FINDA was only briefly introduced in the INP measurements of hailstones in China in Ren et al. (2024), which is not an instrumentation article, we did not provide

a detailed comparison of the original FINDA and FINDA-WLU in our manuscript. We referenced the original version of FINDA in the introduction: "*In this study, we present the newly developed Freezing Ice Nucleation Detector Array at Westlake University (FINDA-WLU), building on the original version of FINDA briefly introduced in Ren et al. (2024).*"

While "user-friendly software" is mentioned, details on real-time monitoring, automated droplet tracking, or data processing algorithms are lacking.

The real-time monitoring is mentioned in Lines 130-132:" *A customized National Instruments LabVIEW program was developed to control the experiment via a user interface panel shown in Fig. 2, including controlling the coolant bath circulator and monitoring the freezing status of droplets in the PCR plate with a CCD camera.*"

We added the following information in Sect. 2.2, Software control.

"*The program can analyze the temporal resolution of grayscale values and determine the frozen temperature of each droplet (details in Sect. 2.3). The frozen fraction and INP number concentration (for both water and air filter samples) are then calculated based on input sample information (calculation methods in Sect. 4.5).*"

The study asserts high precision but omits comparisons with other DFTs (e.g., number of droplets processed per run, false-positive rates). It is recommended to contrast FINDA-WLU directly with existing DFTs in a table, highlighting metrics like droplet capacity, temperature resolution, and uncertainty.

Thank you for your valuable suggestions. We concur with your view that a table summarizing the current DFTs would indeed be very beneficial for readers. However, it is important to note that Miller et al. (2021) have already provided such a table, specifically Table 1.

To address your concern, we have organized a new table based on the one from Miller et al. (2021), which incorporated additional information—such as temperature cooling rate, temperature uncertainty, and $T_{50}$ of water background. Nevertheless, given that Miller et al. (2021) have already presented the majority of the information, we have chosen not to include this table in the manuscript unless the reviewers strongly recommend otherwise.

Table 1. Comparison of droplet freezing techniques (DFTs).

| Name | Description | Drop Size | Drops | T range (°C) | Cooling rate (K min-1) | T uncertainty (°C) | $T_{50}$ of MilliQ (°C) | Citation |
|---|---|---|---|---|---|---|---|---|
|  | combining microfluidic droplet generation and collection with a Peltier-based cold stage | 83-99 µm; 2 µL add 2 µL oil | 250-500 | to −45 (Peltier) | 1-10 | 0.5 |  | Tarn et al., 2018 |

| | | | | | | | | |
|---|---|---|---|---|---|---|---|---|
| CMU-CS | the Carnegie Mellon University Cold Stage system | ~ 0.1 µL | 30-40 | 10 to −40 | 1 | 0.5 | | Polen et al., 2016 |
| FDF | the combined membrane filter-drop freezing technique | 1±0.1 µL | ~ 40, maximum 130 | to ~ −30 | 1 | 0.4 (µL-NIPI) | ~ −27.5; ~ −30 | Price et al., 2018; Schnell, 1982 |
| µL-NIPI | the microlitre Nucleation by Immersed Particle Instrument | 1±0.025 µL | ~ 40 | 1 to −35 | 1 | 0.4 | ~ −26 | Whale et al., 2015 |
| BINARY | the Bielefeld Ice Nucleation ARraY | 1 µL (0.5-5 µL) | 36 | 5 to −40 | 1 (could be 0.1-10) | 0.3 | | Budke and Koop, 2015 |
| WACIFE | a Grant-Asymptote EF600 cold stage | 1.0±0.1 µL, 60-129 µm | ~ 33 | to −40 | 1, 10 | 0.4 | ~ −26 | Wilson et al., 2015 |
| PKU-INA | PeKing University Ice Nucleation Array | 1 µL | 90 | 0 to −30 | 0.1-10 | 0.4 | ~ −28 | Chen et al., 2018 |
| LINA | Leipzig Ice Nucleation Array | 1 µL | 90 | 5 to −40 (same to BINARY) | 1 | 0.5 | ~ −30 | Chen et al., 2018 |
| | a pyroelectric thermal sensor | 1 µL | | to −30 | 1 | 0.8 | | Cook et al., 2020 |
| FRIDGE-TAU | FRankfurt Ice-nuclei Deposition freezinG Experiment, the Tel Aviv University version | 2 µL | 100-130 | −18 to −27 | 1 | | −24 | Ardon-Dryer et al., 2011 |
| DFCP | the NOAA drop freezing cold plate | 2.5 µL | 100 | to −33 | 1-10 | 0.2 | ~ −30 | Baustian et al., 2010; |
| TINA | the Twin-plate Ice Nucleation Assay | 3 µL (0.1-40 µL) | 192, 768 | −1.5 to −40.15 | 1-10 | 0.2 | ~ −26 | Kunert et al., 2018 |
| | a cold stage in single crystals | 3 µL | | 10 to −30 | 3 | | | Mignani et al., 2019 |
| CRAFT | the Cryogenic Refrigerator Applied to Freezing Test | 5 µL | 49 | 50 to −80 | 1 | 0.2 | ~ −35 | Tobo et al., 2016 |
| FINC | Freezing Ice Nuclei Counter | 5-60 µL | 288 | to −30 | 1 | 0.5 | −25.2 (50µL) | Miller et al., 2021 |
| | flow cell microscopy | 20-22 µL | | to −43.15, −93.15 (230 K, 180 K) | 5 | 0.1 | | Koop et al., 2000 |
| AIS | the Automated Ice Spectrometer | 50 µL | 192 | 15 to −33 | 0.69-0.87 | horizontal 0.3; vertical 0.6 | | Beall et al., 2017 |

| | | | | | | | | |
|---|---|---|---|---|---|---|---|---|
| INSEKT | the Ice Nucleation SpEctrometer of the Karlsruhe Institute of Technology | 50 µL | 32 (192 in total) | 0 to −25.15 (248 K to 268 K) | 0.33 | 0.3 | | Schiebel, 2017(thesis) |
| IR-NIPI | the InfraRed-Nucleation by Immersed Particles Instrument | 50 µL | 96 | to −90 | 1 | 0.9 | ~ −18 to −23 | Harrison, et al., 2018 |
| INDA | Ice Nucleation Droplet Array | 50 µL | 96 | to −30 | 1 | 0.5 | ~ −14 to −16 | Chen et al., 2018 |
| DRINCZ | the DRoplet Ice Nuclei Counter Zurich | 50 µL | 96 | 0 to −30 | 1 | 0.9 (reproducible 0.3; horizontal 0.6) | ~ −22.5 | David et al., 2019 |
| DFT | the droplet freezing technique | 50 µL | 48 | 0 to −30 | 0.67 | 1 | ~ −23 | Gute and Abbatt, 2020 |
| CSU-IS | CSU Ice Spectrometer | 50 µL | 32 | to −30 | 0.33 | | start −25 | Barry et al., 2021 |
| | drop freezing apparatus for filters | 0.1 mL | 108 | to −12 | 0.33 (record frozen per 1 °C) | | | Conen et al., 2012 |
| | a high throughput screening platform involving microplates | 150 µL | 96-768 | 2 to −25 | 0.2 | | | Zaragotas et al., 2016 |
| LINDA | LED-based Ice Nucleation Detection Apparatus | 200 µL (40-400 µL) | 52 | to −15 | 0.4 | 0.2 (repeated) | | Stopelli et al., 2014 |
| MINA | the mono ice nucleation assay | | (PCR) | −5 to −15 | 2 for 12 min | | | Pummer et al., 2015 |
| MOUDI−DFT | the micro-orifice uniform deposit impactor-droplet freezing technique (Chow and Watson, 2007) | 0.056-18 µm | | to −40 | to −40 | 0.3 | | Mason et al., 2015 |
| | droplet freezing technique | 1-40 µm | 200−800 | −15.15 to −30.15 | 0.1 | | | Dymarska et al., 2006 |
| | flow cell microscopy technique for aerosol phase transitions | 7-33µm | 65 | to −103.15 | 2−12 | 1 (0.1 at 0 °C) | | Salcedo et al., 2000 |
| Leeds−NIPI | Nucleation by Immersed Particle Instrument | $10^{-12}$ to $10^{-6}$ L (8 µm to 1.45 mm) | | −6 to −36 | 10 | 0.4 | | O'Sullivan et al., 2014 |

| | | | | | | | | |
|---|---|---|---|---|---|---|---|---|
| | | 10-40 µm | 10−230 | to −45.15 (228 K) | 2.5−10 | 0.6 | −32.35 | Murray et al., 2010; Murray et al., 2011 |
| | | 10-200 µm | | ~ 15.15 to −39.15 | 1−2 | The Peltier element below 220K, <1 | ~ −36.15 | Pummer et al., 2012 |
| | a freezing chip | 20-80 µm, 4-300 pL | ~25 | to −40 | 2 | 0.4 | −37.5 | Häusler et al.. 2018 |
| | an FDCS196 cryostage | ~ 35 µm | ~200 | to −40 | 1 | 0.1 (for TMS 94) | −9 | Weng et al., 2017 |
| WISDOM | The Weizmann Supercooled Droplets Observation on Microarray | 40, 100 µm | 500, 120 | 13.15 to −38.15 (260 K to 235 K) | 0.1−10 | 1 | | Reicher et al., 2018 |
| | (Wright and Petters, 2013; Bigg, 1953) | 50-300 µm | ~100−500 | −4 to −33 | 5 | 1 | | Wright et al., 2013 |
| | the differential scanning calorimeter (DSC) measurements, and the cryo-microscope experiments | ~53-96 µm | a few thousand | to −50 | 1 (from −10°C to lower temperature) | 0.3 | | Riechers et al., 2013 |
| | combining microfluidic droplet generation and collection with a Peltier−based cold stage | 83-99 µm; 2 µL add 2 µL oil | 250−500 | to −45 (Peltier) | 1−10 | 0.5 | | Tarn et al., 2018 |
| SBM | soccer ball model (Niedermeier, 2011, 2014, 2015) | 215±70 pL, 107±14 µm | 1200−1500+ | 126.85 to −196.15 | 0.01−100 | 0.1 (from −40°C to 30°C) | | Peckhaus et al., 2016 |
| | a "store and create" microfluidic device | 6 nL (5.8±0.7 nL) equal to 300±18 µm | 720 | 0 to −33 | 1 | 0.2 | −33.7±0.4 | Brubaker et al., 2020 |

Fig. 7 reveals horizontal temperature gradients on the cold stage. While common in DFTs, this issue significantly impacts INP quantification, as ±0.6°C uncertainty may affect INP concentrations to a large extent. How do these gradients affect the freezing temperature statistics (e.g., broadening of spectra)? The original FINDA used dynamic infrared imaging for calibration; FINDA-WLU's "rigorous temperature calibrations" and final INP concentration calibration require elaboration (e.g., correction algorithms).

To obtain the temperature calibration, we compare the frozen fraction curve after the single-well temperature calibration with that before the calibration for pure water and Snomax samples, as shown in Fig. C1.

[Figure]

Fig C1. Frozen fraction curve of Milli-Q water with and without single-well temperature calibration in solid black and red lines, respectively. Frozen fraction of Snomax sample with and without single-well temperature calibration in dashed black and red lines, respectively.

Actually, the FF difference between with and without single-well calibration is not significant, even lower than the FF uncertainties (method in Sect. 3 INP concentration calibration). As the chiller itself causes the horizontal temperature gradients, we strongly recommend that the DFTs include the single-well temperature calibration.

The correction algorithms were already provided in the original manuscript in Lines 257 to 263 (now Lines 259 to 265).

*"To address the horizontal temperature heterogeneity of the PCR plate, an individual-well calibration approach was conducted. The mean temperature of the four calibrated Pt100 sensors ($T_{C\_mean}$) was used to calibrate the temperature of each PCR well:*

$$T_{C\_Infrared_i} = a_i \times T_{C\_mean} + b_i \quad (i = 1, 2, 3, \dots, 96), \tag{3}$$

*where $T_{C\_Infrared_i}$ is the calibrated temperature of the $i^{th}$ well, and $a_i$ and $b_i$ are the slope and intercept of the regression, respectively. For each PCR well, a standard deviation of Eq. (3) is calculated, and two times the largest standard deviation ($\pm 0.22\ °C$) is treated as the uncertainty for this step."*

Fig. 8 shows Milli-Q water freezing at −22°C to −24°C, differing from the listed literature (−13, −14°C). It should specify droplet volumes (not mentioned) and compare with previous studies. It is recommended to test water with documented ultrapure standards and add comparisons to ≥3 DFT studies, especially for studies using similar droplet volumes and numbers, and temperature ranges.

We already had included the information on the droplet size in the manuscript. In the original version of the manuscript, we stated:" *It is worth noting that the droplets measured by micro-PINGUIN (30 µL) and DRINCZ (50 µL) are smaller than or equal to that (50 µL) used in this study.*"

In the revised version of the manuscript, we include more information about different DFTs for the comparison.

*"It is worth noting that the droplet sizes examined in FINC (5 µL) and Micro-PINGUIN (30 µL) are smaller than in FINDA-WLU, while the droplet size used in other DFTs, such as DRINCZ, is equal to the one in FINDA-WLU."*

Also, we now compare water background measurements of more DFT studies, including FINC (Miller et al., 2021), Micro-PINGUIN (Wieber et al., 2024), DRINCZ (David et al., 2019), IR-NIPI (Harrison et al., 2018), and INDA (Chen et al., 2018) to our data (see Figure 8). In particular, the droplet sizes measured by different studies are indicated in the updated Figure (Figure 8).

Below is the updated Figure 8. Our Milli-Q water background (denoted by solid lines) is still one of the lowest among the above-mentioned studies. We changed the main text accordingly.

*"The FF of Milli-Q water droplets using DFTs with different volumes, including Freezing Ice Nuclei Counter (FINC) (Miller et al., 2021), microtiter plate-based ice nucleation detection results in gallium (Micro-PINGUIN) (Wieber et al., 2024), Droplet Ice Nuclei Counter Zurich (DRINCZ) (David et al., 2019), InfraRed-Nucleation by Immersed Particles Instrument (IR-NIPI) (Harrison et al., 2018), and Ice Nucleation Droplet Array (INDA) (Chen et al., 2018), are shown in Fig. 8 for comparison. In general, FINDA-WLU ($T_{50}$ = –26.5 ± 0.04 ℃) shows a considerably lower $T_{50}$ compared to those measured by INDA ($T_{50}$ = –25.5 ℃), FINC ($T_{50}$ = –25.4 ℃), DRINCZ ($T_{50}$ = –22.2 ℃), IR-NIPI ($T_{50}$ = –21.0 ℃), and Micro-PINGUIN ($T_{50}$ = –20.8 ℃)."*

[Figure]

*"Figure 8: Frozen fraction of Milli-Q water. The results of FINDA-WLU are shown as solid lines. The shaded area indicate the measurement uncertainties. Results for other droplet freezing techniques, including FINC (Miller et al., 2021), Micro-PINGUIN (Wieber et al., 2024), DRINCZ (David et al., 2019), IR-NIPI (Harrison et al., 2018), and INDA (Chen et al., 2018), are shown as triangles, squares, dots, and circles, respectively."*

High-concentration dust suspensions (e.g., −2°C onset) in Fig. 9 likely do not reflect atmospheric conditions (typical INP onset: <−15°C). In high concentration suspensions, multiple INPs compete, altering freezing kinetics. Which curves are similar to real atmospheric conditions? Precipitation samples are mentioned but not linked to dust or biological results. Do these samples exhibit similar freezing behavior?

We agree that the high-concentration dust suspensions (e.g., −2°C onset) in Fig. 9 does not reflect the particle concentration in a cloud droplet under atmospheric-relevant conditions. The use of different suspension concentrations is to validate the performance of the FINDA-WLU over a broader temperature range, as droplets with higher particle concentration tend to freeze at a higher temperature. The ability of DFTs to capture ice nucleation events at higher temperatures is important to quantify INP species that are highly ice efficient but exist at low concentrations in the atmosphere.

Regarding atmospheric relevance, it is worth noting that Fig. 9 provides $n_m$ data, which normalizes the freezing ability of the examined sample by particle mass, making $n_m$ independent of particle mass concentrations in single droplets. When applying the conversion by Vali (1971) and to obtain INP concentrations, the assumption of a Poisson distribution of INPs in the droplets is made. This corrects for multiple INPs in single droplets. Overall, $n_m$ is atmospherically revelant and it can estimate the INP number produced by dust particles as long as the particle mass is known. This, as you say, is only true as long as competition for water does not play a role. However, given the comparably large water volume in the examined droplets, we assume that such a competiton does not occur.

A bump at the temperature above −20 °C indicates a contribution of bioaerosol, as observed in our precipitation samples (Fig. 11). This aligns with the findings from many other previous studies (Conen et al., 2011). However, to verify the presence of biological or dust INPs and quantify their contributions, further experiments such as chemical and biological characterizations and heating treatments of samples are needed. As we do not have leftovers of our precipitation samples to do more experiments, and the source characterization of these precipitation samples is not the purpose of this study, no further discussion is added to the original manuscript.

In the manuscript, we stated *"The high INP concentrations in cloud water, snow, and hail at warm temperatures (>−18.0 °C) suggest that biological aerosols might make a great contribution to INPs. Further chemical analysis and heating treatments of samples will help in the future to confirm the nature and sources of INPs."*

It is recommended to include error bars in INP spectra, e.g., Figs. 8–9, to reflect uncertainty.

We totally agree that the error bars (more precisely, the uncertainties) should be included in the results, and we have included the uncertainties in this study, as indicated by error bars in Figs. 8

and 9. But previous studies are lacking this information, which is why we don't have error bars for previous studies in the Figs. 8 and 9.

**Reference:**

Chen, J., Wu, Z., Augustin-Bauditz, S., Grawe, S., Hartmann, M., Pei, X., Liu, Z., Ji, D., and Wex, H.: Ice-nucleating particle concentrations unaffected by urban air pollution in Beijing, China, Atmospheric Chemistry and Physics, 18, 3523-3539, 10.5194/acp-18-3523-2018, 2018.

Conen, F., Morris, C. E., Leifeld, J., Yakutin, M. V., and Alewell, C.: Biological residues define the ice nucleation properties of soil dust, Atmos. Chem. Phys., 11, 9643-9648, 10.5194/acp-11-9643-2011, 2011.

David, R. O., Cascajo-Castresana, M., Brennan, K. P., Rösch, M., Els, N., Werz, J., Weichlinger, V., Boynton, L. S., Bogler, S., Borduas-Dedekind, N., Marcolli, C., and Kanji, Z. A.: Development of the DRoplet Ice Nuclei Counter Zurich (DRINCZ): validation and application to field-collected snow samples, Atmos. Meas. Tech., 12, 6865-6888, 10.5194/amt-12-6865-2019, 2019.

Harrison, A. D., Whale, T. F., Rutledge, R., Lamb, S., Tarn, M. D., Porter, G. C. E., Adams, M. P., McQuaid, J. B., Morris, G. J., and Murray, B. J.: An instrument for quantifying heterogeneous ice nucleation in multiwell plates using infrared emissions to detect freezing, Atmospheric Measurement Techniques, 11, 5629-5641, 10.5194/amt-11-5629-2018, 2018.

Miller, A. J., Brennan, K. P., Mignani, C., Wieder, J., David, R. O., and Borduas-Dedekind, N.: Development of the drop Freezing Ice Nuclei Counter (FINC), intercomparison of droplet freezing techniques, and use of soluble lignin as an atmospheric ice nucleation standard, Atmos. Meas. Tech., 14, 3131-3151, 10.5194/amt-14-3131-2021, 2021.

Ren, Y., Fu, S., Bi, K., Zhang, H., Lin, X., Cao, K., Zhang, Q., and Xue, H.: Freezing Nucleus Spectra for Hailstone Samples in China From Droplet Freezing Experiments, Journal of Geophysical Research: Atmospheres, 129, e2023JD040505, 10.1029/2023JD040505, 2024.

Vali, G.: Quantitative evaluation of experimental results an the heterogeneous freezing nucleation of supercooled liquids, Journal of Atmospheric Sciences, 28, 402-409, 1971.

Wieber, C., Rosenhøj Jeppesen, M., Finster, K., Melvad, C., and Šantl-Temkiv, T.: Micro-PINGUIN: microtiter-plate-based instrument for ice nucleation detection in gallium with an infrared camera, Atmos. Meas. Tech., 17, 2707-2719, 10.5194/amt-17-2707-2024, 2024.

---

## Author Comment (AC2)

**Reviewer #2**

General comments:

This paper describes an improved droplet freezing instrument and gives examples of its performance. The authors took care to account and correct for temperature gradients within the plate by applying a rigorous temperature calibration. Like this, they achieve a temperature uncertainty of $\pm$ 0.6°C. Moreover, they developed user-friendly software with automatic freezing detection.

We would like to thank the reviewers for their thoughtful comments that helped improve our manuscript. We revised the manuscript accordingly and think it has strengthened as a result. Please find our point-by-point response in blue text. Additions to the text are shown in *italics with an underline*. All line numbers refer to the new version of the draft. A tracked changes version is also included.

To achieve the low temperature uncertainty of $\pm$ 0.6°C, the temperature of each individual well is measured with an infrared camera. These measurements show a temperature increase in two steps due to the heat release during freezing over a temperature decrease of the ethanol bath by about 1°C. The authors assign ice nucleation to the first heat release without explaining why. Yet, to achieve the high accuracy of $\pm$ 0.6°C, the correct detection of the instance of ice nucleation is crucial. If the exact instance of ice nucleation is not identified, this will add to the temperature uncertainty.

Thank you for your comment. We agree that detecting the instance of ice nucleation is critically important. This comment is related to your later specific comment in Lines 153–155 and Figure 3.

Figure 3 demonstrates a decrease in grayscale as the chamber cools. The figure should be interpreted from right to left, as the experiment represents a cooling process. Consequently, the first significant change in grayscale indicates the onset of ice nucleation. The second change in grayscale is caused by the freezing of the remaining droplet solution (David et al., 2019). We have also checked previously published papers, and most figures depict the temperature decrease from right to left. Therefore, we have decided to retain the figure as it is.

We modified the text accordingly: "*The grayscale value of a well stays constant until a sudden decrease is observed during a cooling experiment, indicating the onset of freezing. From 0.0 °C to −35.0 °C, the maximum decrease in grayscale value was used to identify the freezing event and the temperature at which it occurs.*"

Moreover, according to the infrared camera measured temperature during a cooling experiment (Fig. C1), only one latent heat release process happened. Similar to Fig. 5d in the manuscript, here we show the temperature profile of a well during the cooling process.

[Figure]

Fig. C1. The temperature profile of a well is measured by the infrared camera after its calibration.

To account for horizontal temperature differences within the well plate, a temperature correction for each well is performed. Such a correction requires that the wells' temperature deviations from the average of the thermocouples is highly reproducible. The authors need to evaluate this potential contribution to temperature uncertainty.

The horizontal temperature heterogeneity is caused by the ethanol circulation in the chiller, which has been discussed in previous studies, e.g., the DRoplet Ice Nuclei Counter Zurich (DRINCZ) (David et al., 2019), IR-NIPI (Harrison et al., 2018), and Micro-PINGUIN (Wieber et al., 2024). For a specific chiller and fixed temperature cooling rate, the temperature deviation is reproducible.

The horizontal temperature distribution might slightly change over time. To avoid a bias in the temperature calibration, we plan to conduct the whole temperature calibration procedure yearly.

Moreover, the freezing temperature measured for pure water should be compared to additional instruments.

Thanks for your suggestions. We compared with more DFT studies, including FINC (Miller et al., 2021), Micro-PINGUIN (Wieber et al., 2024), DRINCZ (David et al., 2019), IR-NIPI (Harrison et al., 2018), and INDA (Chen et al., 2018b).

Below is the updated Figure 8. Our Milli-Q water background (denoted by solid lines) is still one of the lowest among the above-mentioned studies. We changed the main text accordingly.

*"The FF of Milli-Q water droplets using DFTs with different volumes, including Freezing Ice Nuclei Counter (FINC) (Miller et al., 2021), microtiter plate-based ice nucleation detection results in gallium (Micro-PINGUIN) (Wieber et al., 2024), Droplet Ice Nuclei Counter Zurich (DRINCZ) (David et al., 2019), InfraRed-Nucleation by Immersed Particles Instrument (IR-NIPI) (Harrison et al., 2018), and Ice Nucleation Droplet Array (INDA) (Chen et al., 2018b), are shown in Fig. 8 for comparison. In general, FINDA-WLU ($T_{50} = -26.5 \pm 0.04°C$) shows a considerably lower $T_{50}$ compared to those measured by INDA ($T_{50} = -25.5\,°C$), FINC ($T_{50} = -25.4\,°C$), DRINCZ ($T_{50} = -22.2\,°C$), IR-NIPI ($T_{50} = -21.0\,°C$), and Micro-PINGUIN ($T_{50} = -20.8\,°C$)."*

[Figure]

*"Figure 8: Frozen fraction of Milli-Q water. The results of FINDA-WLU are shown as solid lines. The shaded area indicate the measurement uncertainties. Results for other droplet freezing techniques, including FINC (Miller et al., 2021), Micro-PINGUIN (Wieber et al., 2024), DRINCZ (David et al., 2019), IR-NIPI (Harrison et al., 2018), and INDA (Chen et al., 2018b), are shown as triangles, squares, dots, and circles, respectively."*

Overall, the manuscript is well written except for the introduction. Here, the strength and weaknesses of the different immersion freezing setups are not enough pointed out and discussed. The state of the art of freezing instruments does not discriminate sufficiently between different types of setups in terms of temperature range that is accessible and how the covered sample volume depends on the droplet preparation technique. Moreover, the references given in the introduction are not sufficiently balanced (see specific comments).

Thanks for your suggestion. As you mentioned, Miller et al. (2021) summarize different DFTs. Below, we have incorporated additional information—such as temperature cooling rate, temperature uncertainty, and $T_{50}$ of water background—into a new table that is based on the original Table 1 from Miller et al. (2021). Nevertheless, given that Miller et al. (2021) have already presented the majority of the pertinent information, we have chosen not to include this table in the manuscript unless the reviewers strongly recommend otherwise.

We did modify the text in the introduction. A more detailed response is given in the specific comment.

Table 1. Comparison of droplet freezing techniques (DFTs).

| Name | Description | Drop Size | Drops | T range (°C) | Cooling rate (K min-1) | T uncertainty (°C) | $T_{50}$ of MilliQ (°C) | Citation |
|---|---|---|---|---|---|---|---|---|
| | | | | | | | | |

| | | | | | | | | |
|---|---|---|---|---|---|---|---|---|
| | combining microfluidic droplet generation and collection with a Peltier-based cold stage | 83-99 μm; 2 μL add 2 μL oil | 250-500 | to −45 (Peltier) | 1-10 | 0.5 | | Tarn et al., 2018 |
| CMU-CS | the Carnegie Mellon University Cold Stage system | ~ 0.1 μL | 30-40 | 10 to −40 | 1 | 0.5 | | Polen et al., 2016 |
| FDF | the combined membrane filter-drop freezing technique | 1±0.1 μL | ~ 40, maximum 130 | to ~ −30 | 1 | 0.4 (μL-NIPI) | ~ −27.5; ~ −30 | Price et al., 2018; Schnell, 1982 |
| μL-NIPI | the microlitre Nucleation by Immersed Particle Instrument | 1±0.025 μL | ~ 40 | 1 to −35 | 1 | 0.4 | ~ −26 | Whale et al., 2015 |
| BINARY | the Bielefeld Ice Nucleation ARraY | 1 μL (0.5-5 μL) | 36 | 5 to −40 | 1 (could be 0.1-10) | 0.3 | | Budke and Koop, 2015 |
| WACIFE | a Grant-Asymptote EF600 cold stage | 1.0±0.1 μL, 60-129 μm | ~ 33 | to −40 | 1, 10 | 0.4 | ~ −26 | Wilson et al., 2015 |
| PKU-INA | PeKing University Ice Nucleation Array | 1 μL | 90 | 0 to −30 | 0.1-10 | 0.4 | ~ −28 | Chen et al., 2018 |
| LINA | Leipzig Ice Nucleation Array | 1 μL | 90 | 5 to −40 (same to BINARY) | 1 | 0.5 | ~ −30 | Chen et al., 2018 |
| | a pyroelectric thermal sensor | 1 μL | | to −30 | 1 | 0.8 | | Cook et al., 2020 |
| FRIDGE-TAU | FRankfurt Ice-nuclei Deposition freezinG Experiment, the Tel Aviv University version | 2 μL | 100-130 | −18 to −27 | 1 | | −24 | Ardon-Dryer et al., 2011 |
| DFCP | the NOAA drop freezing cold plate | 2.5 μL | 100 | to −33 | 1-10 | 0.2 | ~ −30 | Baustian et al., 2010; |
| TINA | the Twin-plate Ice Nucleation Assay | 3 μL (0.1-40 μL) | 192, 768 | −1.5 to −40.15 | 1-10 | 0.2 | ~ −26 | Kunert et al., 2018 |
| | a cold stage in single crystals | 3 μL | | 10 to −30 | 3 | | | Mignani et al., 2019 |
| CRAFT | the Cryogenic Refrigerator Applied to Freezing Test | 5 μL | 49 | 50 to −80 | 1 | 0.2 | ~ −35 | Tobo et al., 2016 |
| FINC | Freezing Ice Nuclei Counter | 5-60 μL | 288 | to −30 | 1 | 0.5 | −25.2 (50μL) | Miller et al., 2021 |

| | | | | | | | | |
|---|---|---|---|---|---|---|---|---|
| | flow cell microscopy | 20-22 µL | | to −43.15, −93.15 (230 K, 180 K) | 5 | 0.1 | | Koop et al., 2000 |
| AIS | the Automated Ice Spectrometer | 50 µL | 192 | 15 to −33 | 0.69-0.87 | horizontal 0.3; vertical 0.6 | | Beall et al., 2017 |
| INSEKT | the Ice Nucleation SpEctrometer of the Karlsruhe Institute of Technology | 50 µL | 32 (192 in total) | 0 to −25.15 (248 K to 268 K) | 0.33 | 0.3 | | Schiebel, 2017(thesis) |
| IR-NIPI | the InfraRed-Nucleation by Immersed Particles Instrument | 50 µL | 96 | to −90 | 1 | 0.9 | ~ −18 to −23 | Harrison, et al., 2018 |
| INDA | Ice Nucleation Droplet Array | 50 µL | 96 | to −30 | 1 | 0.5 | ~ −14 to −16 | Chen et al., 2018 |
| DRINCZ | the DRoplet Ice Nuclei Counter Zurich | 50 µL | 96 | 0 to −30 | 1 | 0.9 (reproducible 0.3; horizontal 0.6) | ~ −22.5 | David et al., 2019 |
| DFT | the droplet freezing technique | 50 µL | 48 | 0 to −30 | 0.67 | 1 | ~ −23 | Gute and Abbatt, 2020 |
| CSU-IS | CSU Ice Spectrometer | 50 µL | 32 | to −30 | 0.33 | | start −25 | Barry et al., 2021 |
| | drop freezing apparatus for filters | 0.1 mL | 108 | to −12 | 0.33 (record frozen per 1 °C) | | | Conen et al., 2012 |
| | a high throughput screening platform involving microplates | 150 µL | 96-768 | 2 to −25 | 0.2 | | | Zaragotas et al., 2016 |
| LINDA | LED-based Ice Nucleation Detection Apparatus | 200 µL (40-400 µL) | 52 | to −15 | 0.4 | 0.2 (repeated) | | Stopelli et al., 2014 |
| MINA | the mono ice nucleation assay | | (PCR) | −5 to −15 | 2 for 12 min | | | Pummer et al., 2015 |
| MOUDI−DFT | the micro-orifice uniform deposit impactor-droplet freezing technique (Chow and Watson, 2007) | 0.056-18 µm | | to −40 | to −40 | 0.3 | | Mason et al., 2015 |
| | droplet freezing technique | 1-40 µm | 200−800 | −15.15 to −30.15 | 0.1 | | | Dymarska et al., 2006 |

| | | | | | | | | |
|---|---|---|---|---|---|---|---|---|
| | flow cell microscopy technique for aerosol phase transitions | 7-33μm | 65 | to −103.15 | 2−12 | 1 (0.1 at 0 °C) | | Salcedo et al., 2000 |
| Leeds−NIPI | Nucleation by Immersed Particle Instrument | $10^{-12}$ to $10^{-6}$ L (8 μm to 1.45 mm) | | −6 to −36 | 10 | 0.4 | | O'Sullivan et al., 2014 |
| | | 10-40 μm | 10−230 | to −45.15 (228 K) | 2.5−10 | 0.6 | −32.35 | Murray et al., 2010; Murray et al., 2011 |
| | | 10-200 μm | | ~ 15.15 to −39.15 | 1−2 | The Peltier element below 220K, <1 | ~ −36.15 | Pummer et al., 2012 |
| | a freezing chip | 20-80 μm, 4-300 pL | ~25 | to −40 | 2 | 0.4 | −37.5 | Häusler et al.. 2018 |
| | an FDCS196 cryostage | ~ 35 μm | ~200 | to −40 | 1 | 0.1 (for TMS 94) | −9 | Weng et al., 2017 |
| WISDOM | The WeIzmann Supercooled Droplets Observation on Microarray | 40, 100 μm | 500, 120 | 13.15 to −38.15 (260 K to 235 K) | 0.1−10 | 1 | | Reicher et al., 2018 |
| | (Wright and Petters, 2013; Bigg, 1953) | 50-300 μm | ~100−500 | −4 to −33 | 5 | 1 | | Wright et al., 2013 |
| | the differential scanning calorimeter (DSC) measurements, and the cryo-microscope experiments | ~53-96 μm | a few thousand | to −50 | 1 (from −10°C to lower temperature) | 0.3 | | Riechers et al., 2013 |
| | combining microfluidic droplet generation and collection with a Peltier−based cold stage | 83-99 μm; 2 μL add 2 μL oil | 250−500 | to −45 (Peltier) | 1−10 | 0.5 | | Tarn et al., 2018 |
| SBM | soccer ball model (Niedermeier, 2011, 2014, 2015) | 215±70 pL, 107±14 μm | 1200−1500+ | 126.85 to −196.15 | 0.01−100 | 0.1 (from −40°C to 30°C) | | Peckhaus et al., 2016 |
| | a "store and create" microfluidic device | 6 nL (5.8±0.7 nL) equal to 300±18 μm | 720 | 0 to −33 | 1 | 0.2 | −33.7±0.4 | Brubaker et al., 2020 |

Specific comments:

In the title, the abstract, and in the text, the impression is given that the FINDA-WLU is based on a previous design that has been improved. Yet, no reference to the previous design is given. Please explain.

The first generation of FINDA was designed in 2021 by Kai Bi (one of our corresponding authors) from the Beijing Weather Modification Center. The original version FINDA was used to measure the INP of hailstones in China (details in Ren et al. (2024)). The new version was redesigned in cooperation with Westlake University. We updated the setup, hardware, and temperature calibration procedure for the version of FINDA-WLU.

[Figure]

Fig. 2 in Ren et al. (2024) shows the original verion of FINDA.

Based on your suggestion, we modified the introduction to include this information. "*In this study, we present the newly developed Freezing Ice Nucleation Detector Array at Westlake University (FINDA-WLU), building on the original version of FINDA briefly introduced in Ren et al. (2024).*"

Lines 50–54: The references given for in-situ methods and laminar flow reactors are not sufficiently balanced and seem to have a bias to references from authors of the manuscript. Specific examples of ice nucleation chambers, laminar flow reactors, and droplet freezing devices should be given together with appropriate references. See Miller et al., 2021 for an overview of instruments.

Thanks for your suggestion.

Regarding the ice nucleation chambers and laminar flow reactors, we cited the CSU-CFDC (Rogers, 1988; Rogers et al., 2001; Demott et al., 2015), SPIN (Garimella et al., 2016), HINC (Lacher et al., 2017), PINC (Kanji et al., 2013), and PINE (Möhler et al., 2021a). Regarding the offline DFTs, we cited the CUS-IS (Hill et al., 2014), BINARY (Budke and Koop, 2015), FINC (Miller et al., 2021), PKU-INA (Chen et al., 2018a), INDA (Chen et al., 2018b), LINA (Chen et al., 2018b), IR-NIPI (Harrison et al., 2018), DRINCZ (David et al., 2019), and INSEKT (Steinke et al., 2020).

We modified the introduction accordingly, including the above-mentioned instrument papers.

Lines 61–64: "However, ice nucleation chambers and reactors are typically expensive and have higher detection limitations compared to DFTs, especially at higher temperatures ($T > –20°C$) where increased background noise caused by ice residues falling from chamber walls or counting statistics of low ice crystal numbers makes detecting INPs with low concentrations challenging." What is meant by "higher detection limitations"? To my knowledge, ice-nucleation chambers do not have a problem with falling ice. Please give references for this statement.

Thank you for pointing that out. Continuous Flow Diffusion Chambers (CFDCs) indeed suffer from the falling ice issue. We quote from Lacher et al. (2017), which explains the working principle of the Horizontal Ice Nucleation Chamber (HINC), a typical CFDC designed at ETH.

"*During an ice nucleation experiment, erroneous counts in the OPC ice channel can arise from electrical noise in the OPC or from internal ice sources such as frost falling off the warmer chamber wall giving rise to particle counts that are falsely classified as ice.*"

While an expansion chamber (e.g., PINE) does not encounter the falling ice issue, it can still produce erroneous counts in the OPC at warmer temperatures, especially when INP concentrations in the air are low. And generally, the optical detection of single frozen droplets as used in ice-nucleation chambers requires much higher INP concentrations, which is what we ultimately mean by higher detection limits.

To clarify, we have revised the text to "*However, ice nucleation chambers and reactors are typically expensive and have higher detection limits compared to DFTs. This enables them to measure often only at lower temperatures (T < –20 °C), particularly for typical atmospheric INP concentrations. Background noise caused by ice residues falling from chamber walls (e.g., CFDCs) or counting statistics of low ice crystal numbers make detecting INPs with low concentrations challenging (e.g., for both CFDCs and expansion chambers).*"

Lines 65–69: The references cited here are mostly about measurement campaigns and do not give detailed instrument descriptions. Moreover, they are all given as one list. Instead, they should be split up into microliter and picoliter setups, and into microfluidic devices and instruments working with well plates. References about measurement campaigns need to be replaced by references describing the instrument setup.

Thank you for your suggestion. In response to your previous comments, we have revised the cited references to include more classical and instrumental sources.

As for Lines 66–70, the revised text now reads: "*As an alternative, offline DFTs have been developed to measure the temperature-dependent freezing abilities of droplets containing aerosol particles. While different DFTs follow similar principles, the methods may differ for sample collection, droplet preparation, and sample cooling (Hill et al., 2014; Budke and Koop, 2015; Chen et al., 2018a; Miller et al., 2021; Chen et al., 2018b; Harrison et al., 2018; David et al., 2019; Steinke et al., 2020).*"

When comparing the DFTs' performance of MilliQ water samples, we include the droplet size information (Fig. 8 in the revised manuscript). Moreover, when summarizing the DFT instruments, we also include the droplet size information. However, we chose not to separate the microliter and picoliter setups, as we did not specifically discuss their common features or differences. We believe that dividing them would disrupt the logical flow of this section.

Lines 71–74: "Typically, the sampling time, droplet volume, and aerosol suspension concentration can be adjusted, which affects the particle number within each droplet and, thereby, its freezing ability. For example, particle numbers within a droplet can be enhanced by extending the aerosol sampling time, enlarging the droplet size, or reducing the dilution ratio of aerosol suspensions with water.": The possibilities of adjustment that are pointed out here are typically small, because most instruments can work only in a narrow volume range (within less than an order of magnitude). Variations in sampling time are also within a quite narrow range. Droplet experiments are usually performed with a cooling rate of 1 K/min because at higher cooling rates the temperature accuracy decreases and experiments at lower cooling rates become time consuming. The authors need to demonstrate the volume and cooling rate range that they can cover with their setup.

Thanks for your suggestions. In this study, we used a fixed cooling rate of 1 K min$^{-1}$ and a droplet volume of 50 μL.

FINDA-WLU, similar to other DFTs, can change the droplet volume (a relatively narrow range) and the cooling rate range. However, the volume and cooling rate range may impact the temperature uncertainty, which means a companion calibration should be provided. We would perform such a calibration if experiments with different droplet sizes and cooling rates are needed.

We agree that the possibilities of adjusting droplet size for FINDA are small. But the aerosol sampling time and aerosol suspensions can be adjusted for a large range, e.g., we adjust the solution by about 2 and 9 orders of magnitude for ATD and Snomax® solutions. Also, if microfluidic chips are used for droplet generation, the volume can be largely modulated.

Lines 75–77: "In this way, this approach enables the quantification of low-concentration INP species in the atmosphere, which overcomes the high detection limitations of ice nucleation chambers. Due to these advantages, DFTs are widely used in current ice nucleation studies." DFTs operating with well plates are widely used because they are rather cheap and easy to use. Instruments working with smaller volumes like microfluidic devices and continuous flow diffusion chambers are complementary to well plate setups because they can monitor ice nucleation down to the homogeneous freezing threshold while setups with well plates only deliver results down to temperatures where freezing on "pure water" impurities sets in, which is well above the homogeneous freezing threshold. The limitations of the FINDA setup should be pointed out clearly. The temperature ranges covered with the different setups should be discussed.

We agree with you that (1) DFTs operating with well plates are widely used because they are rather cheap and easy to use; (2) Instruments working with smaller volumes, like microfluidic devices and continuous flow diffusion chambers, are complementary to well plate setups because they can monitor ice nucleation down to the homogeneous freezing threshold.

In lines 57-61, we include the above discussion:" *To measure the immersion freezing of droplets containing INPs, ice nucleation chambers are operated under mixed-phase cloud-relevant conditions, with T above –38 ℃ and RH with respect to water at ~100%. The continuous flow diffusion chambers (CFDCs) (Demott et al., 2017; Lacher et al., 2017; Demott et al., 2018; Brunner and Kanji, 2021) and cloud expansion chambers (Möhler et al., 2021a; Möhler et al., 2021b) are two types of ice nucleation chambers operating on different working principles.*"

We also modified the text in Lines 61 to 65.

*"However, ice nucleation chambers and reactors are typically expensive and have higher detection limits compared to DFTs. This enables them to measure often only at lower temperatures (T < –20 ℃), particularly for typical atmospheric INP concentrations. Background noise caused by ice residues falling from chamber walls (e.g., CFDCs) or counting statistics of low ice crystal numbers make detecting INPs with low concentrations challenging (e.g., for both CFDCs and expansion chambers).*"

Lines 101–102: "FINDA-WLU detects LED light reflected by freezing of water droplets placed in a 96-well PCR plate over time." Sentence needs to be improved.

The full sentence is "*Using a CCD camera (Fig. 1a), FINDA-WLU detects LED light reflected by freezing of water droplets placed in a 96-well PCR plate over time.*"

It was changed to: "*A CCD camera (Fig. 1a) is used to detect the reflected LED light over the water droplets placed in a 96-well PCR plate during the experiment.*"

Lines 103–104: "The camera is fixed above the PCR wells region using an adjustable camera zoom lens (12-120 mm Focal Length, Qiyun Photoelectric Co., China)." Sentence structure needs to be improved.

It was changed to: "*The camera is fixed above the region of the PCR wells using an adjustable zoom lens (12-120 mm Focal Length, Qiyun Photoelectric Co., China).*"

Lines 114–116: "These sensors are embedded and sealed within thermally conductive epoxy (Omegabond 200, Omega Engineering, Inc., USA) within tubes cut from a PCR plate, ensuring consistent heat transfer between the PCR plate and Pt100 sensors." How are the tubes cut? Does this mean that the commercial plates are modified?

We cut a well from the PCR and put the temperature sensor in the well, with thermally conductive

epoxy. Below is the figure of this setup.

[Figure]

Fig C2. (a) The modified Pt100 sensors. (b) The modified Pt100 sensors with thermally conductive epoxy inside a PCR well.

The cut well is outside of the PCR plate, as shown in Fig. 1b and c. Therefore, the commercial plates are not modified in each experiment.

Lines 153–155: Figure 3 shows an increase in grayscale not a decrease. Please revise the text accordingly. Moreover, Fig. 3 shows that a large change in grayscale (by about 80) is always preceded by a smaller change by around 20 at about 1 K higher temperature. What makes you sure that the second larger change marks ice nucleation and not the smaller one at higher temperature? As the accuracy of the instrument is given as ± 0.6 K, it is important whether the first small or the second larger step marks nucleation. This needs to be investigated and discussed.

Figure 3 demonstrates a decrease in grayscale as the chamber cools. The figure should be interpreted from right to left, as the experiment represents a cooling process. Consequently, the first significant change in grayscale indicates the onset of ice nucleation. The second change in grayscale is caused by the freezing of the remaining droplet solution (David et al., 2019). We have also checked previously published papers, and most figures depict the temperature decrease from right to left. Therefore, we have decided to retain the figure as it is.

We modified the text accordingly: "*The grayscale value of a well stays constant until a sudden decrease is observed during a cooling experiment, indicating the onset of freezing. From 0.0 °C to −35.0 °C, the maximum decrease in grayscale value was used to identify the freezing event and the temperature at which it occurs.*"

Moreover, according to the infrared camera measured temperature during a cooling experiment (Fig. C1), only one latent heat release process happened. Similar to Fig. 5d in the manuscript, here we show the temperature profile of a well during the cooling process.

[Figure]

Fig. C1. The temperature profile of a well is measured by the infrared camera after its calibration.

Lines 235–236: "This phenomenon also explains why freezing is most often triggered at the droplet bottom from our observation." What observation do you refer to? Can you observe where freezing starts? Also, the temperature difference of just 1°C between the bottom and the top of the well is not sufficient to trigger freezing always from the bottom, especially when freezing occurs over a large temperature range.

In our experiment, we observed that ice nucleation began at the bottom of the PCR well, not only for FINDA-WLU but also for other PCR well-based cold stages. However, this observation cannot be demonstrated through the figures. Therefore, we have removed the sentence to avoid any ambiguity.

Lines 119–120, line 245, Figure 6: The figure shows that almost 5°C are required until the temperature difference becomes linear. As samples may freeze already at around -5°C, consider to starting the ramp at 5°C so that a good linearity is achieved when temperature reaches subzero temperatures. Just one cooling ramp is shown in Fig. 6. Have the cooling ramps been repeated? What is the reproducibility?

This experiment was repeated multiple times with multiple well positions in a PCR, and it is reproducible. We agree with you that starting the freezing from 5 °C will solve this problem.

Importantly, the purpose of this test in Fig. 6 is to verify which temperatures (bottom of the aluminum block, bottom of the empty PCR plate, Milli-Q water surface, and ethanol surface) should be used for horizontal temperature calibration. Section 2.4.3 and Figure 6 do not include the temperature calibration; therefore, the non-linear correlation above −5 °C will not affect the calibration results.

Line 250: David et al. (2019) does not use an aluminium block.

In the original text, we stated:" *The temperature bias across 96-well PCR plates has been discussed for aluminum block-based instruments with simulations (Beall et al., 2017), calibration substance freezing experiments (Kunert et al., 2018), and by comparison of temperature differences between corner and center wells (David et al., 2019).*"

We are trying to say that David et al. (2019) compares the temperature differences between the corner and center wells. We did not mean it use the aluminum block-based instruments, but Beall et al. (2017) use the aluminum block-based instrument.

Line 265, Figure 7: how many times has the well calibration experiment been performed? What was the variability between experiments? Has it been performed with different PCR plates? There might be additional variability introduced when the position of the plates within the block has some variability.

We performed the single-well calibration experiment only once. As we responded in the previous comments, the horizontal temperature heterogeneity is caused by the ethanol circulation in the chiller, which has been discussed in previous studies, e.g., the DRoplet Ice Nuclei Counter Zurich (DRINCZ) (David et al., 2019), IR-NIPI (Harrison et al., 2018), and Micro-PINGUIN (Wieber et al., 2024). For a specific chiller and fixed temperature cooling rate, we assume the temperature deviation is reproducible.

The horizontal temperature distribution might slightly change over time. To avoid the bias of temperature calibration, we do the whole temperature calibration procedure yearly.

We used PCR plates from the same brand, as different brands may have varying thermal conductivities. Therefore, when using PCR plates from different brands, additional temperature calibration is required.

The aluminum block is fixed inside the chiller, ensuring that the PCR position remains consistent across all experiments.

Line 333: The method by Agresti and Coull (1998) should be described in some sentences.

It was explained in Lines 281-284.

"*$C_{INP}(T)$ is calculated from statistical analysis; therefore, it is necessary to assess the reliability of the results. According to the binomial distribution method proposed by Agresti and Coull (1998), the 95% confidence interval of the FF at temperature T, $CI_{95\%}(T)$, is calculated as…*"

Line 346, Figure 9: Can you specify what kind of uncertainty the shaded area shows? Min-max or percentiles? How many times was a measurement repeated?

The corresponding shaded areas indicate the 95% confidence interval of $n_m$, derived by Agresti and Coull (1998). We modified the figure caption accordingly.

Line 355: a reference to Fig. A2 in the appendix would be helpful here.

As all FF of Snomax are from this study, a reference to Fig. A2 is not needed.

Line 369: References to the "previous studies" should be given.

Done. We added previous studies (Wieber et al., 2024; Tarn et al., 2018; Polen et al., 2016).

Line 373, Figure 10: the figure caption needs to be reformulated. Moreover, the references to the studies should be added. The freezing experiments seem to have been carried out several times as uncertainty ranges are indicated in the figures. It needs to be stated how many times.

The references are added.

We only did one experiment for each dilution of the Snomax® samples. The uncertainty range is the 95% confidence interval of $n_{\mathrm{m}}$, derived by Agresti and Coull (1998).

Technical comments:

Line 220: "bottom of the" instead of "bottom of"

Changed.

Line 332: "Fig. 6" should be "Fig. 9"

Changed.

Line 335: "overlapping" instead of "overlapped"

Changed.

Line 351: "bacteria" instead of "bacteriuma"

Changed.

Line 356: "scale" instead of "are scaled"

Changed.

Line 370: "Caution" instead of "Cautions"

Changed.

Line 380: "overlapping" instead of "overlapped"

Changed.

Line 381: "who" instead of "which"

Changed.

Reference:

[revised manuscript text omitted]